# UNITED WE TRAIN, DIVIDED WE FAIL! REPRESENTATION LEARNING FOR TIME SERIES BY PRETRAINING FROM 75 DATASETS AT ONCE

## ABSTRACT

In natural language processing and vision, pretraining is utilized to learn effective representations. Unfortunately, the success of pretraining does not easily carry over to time series due to potential mismatch between sources and target. Actually, common belief is that multi-dataset pretraining does not work for time series! Au contraire, we introduce a new self-supervised contrastive pretraining approach to learn one encoding from many unlabeled and diverse time series datasets, so that the single learned representation can then be reused in several target domains for, say, classification. Specifically, we propose the *XD-MixUp* interpolation method and the *Soft Interpolation Contextual Contrasting* (SICC) loss. Empirically, this outperforms both supervised training and other self-supervised pretraining methods when finetuning on low-data regimes. This disproves the common belief: We can actually learn from multiple time series datasets, even from 75 at once.

## 1 INTRODUCTION

The recent success of large language models (Devlin et al., 2019; Brown et al., 2020) as well as diffusion models (Rombach et al., 2022) has shown that leveraging vast amounts of text and image data can dramatically improve the performance of deep learning models. This has led to impressive advancements in several application areas, such as translation, interactive chat assistants, and text-conditioned image generation. Apparently, there is still a need for a methodology to apply the same principles to the time domain, as significant amounts of *unlabeled* time domain data are available, yet they are seldom used for training models for different tasks. Most classification systems on time series are supervised and therefore still rely on expensive and complete labels per dataset (Shi et al., 2021; Nie et al., 2023; Zeng et al., 2023). Overcoming this is crucial in contemporary real-world situations, such as healthcare, where labeled data is not the only limitation. Additionally, there is often a scarcity of sufficient data points, for instance, due to privacy constraints. However, this clashes with the requirements of current deep learning models, which require large single-source datasets (Iwana & Uchida, 2021). Fortunately, there exist multiple small datasets which, in combination, can be leveraged even if they are unlabeled. For instance, the UCR/UEA Time Series Classification Archive (Dau et al., 2019) contains 57% of datasets with 300 or fewer training samples, which are usually not enough to be applicable to the tasks. Additionally, there are datasets with only unlabeled data, such as the M4 Competition (Makridakis et al., 2018) or recordings of meteorological, financial, industrial, traffic, and other signals. Combining them is a new perspective on this collection of valuable data.

We address these challenges by utilizing transfer learning, and specifically, by training a representation on unlabeled source datasets (Eldele et al., 2021). As shown in Figure 1, the learned features are then used as a starting point for finetuning on a target dataset with typically much fewer labeled instances. Although multiple works have shown the feasibility of pretraining on time series (Ma et al., 2023), the success of pretraining does not easily carry over to the time series modality due to a potential mismatch between sources and target. In the image domain, we can leverage pretrained weights from models trained on a general dataset, even in largely different domains. However, for time series, the source and target domain currently need to be quite similar and follow the same underlying temporal dynamics (Zhang et al., 2022). Thus, a general representation taking advantage of a diverse collection of source datasets can prove very beneficial for use in several target domains.

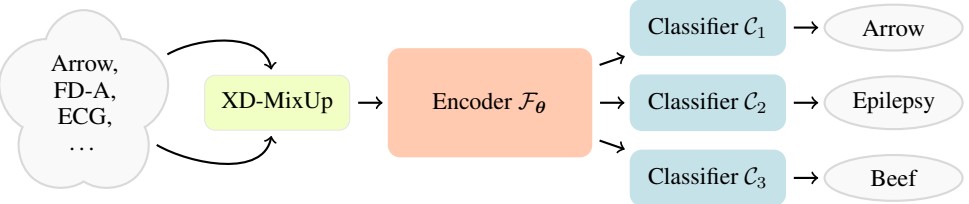

Figure 1: The core idea of our method XIT is to learn a single encoder from multiple datasets. The resulting representation can then be used to train classifiers on datasets seen during the pretraining phase and to be transferred to entirely new ones.

**Contributions** To this end, we make several important contributions: (**1**) We show how up to 75 unlabeled time series datasets can be combined effectively into a single pretraining collection. (**2**) We propose a novel interpolation method **C**ross-**D**ataset MixUp (XD-MixUp) based on (Zhang et al., 2018) that induces a shared latent representation for multiple datasets. (**3**) We propose the **S**oft **I**nterpolation **C**ontextual **C**ontrasting (SICC) loss function, which is incorporated into the Time Series Temporal and Contextual Contrasting (TS-TCC) (Eldele et al., 2021) framework using XD-MixUp. Overall, we call our new architecture XIT[1] (**X**D-MixUp + **S**ICC + **T**emporal Contrasting). (**4**) We demonstrate good transfer classification performance on multiple small labeled target datasets without requiring extensive retraining for each. In particular, we outperform supervised training and other self-supervised pretraining methods.

**Structure of the Paper** We start with explaining our proposed XIT method, including introducing the XD-MixUp and the SICC loss, before moving on to the empirical demonstration of its efficacy. Before concluding, we present the related works.

## 2 MULTI-DATASET PRETRAINING WITH XIT

In this section, we present our pretraining method XIT, where the overall goal is to learn a $D$-dimensional latent representation $z_i \in \mathbb{R}^D$ of some time series $x_i \in \mathbb{R}^T$ of length $T$. For clarity, we focus on univariate time series while the method can be readily extended to multivariate tasks. We train the parameters $\theta$ of an encoder $\mathcal{F}_{\theta}(\cdot)$ to compress $x_i$ into a more abstract representation $z_i = \mathcal{F}_{\theta}(x_i)$. That representation can subsequently be used for downstream tasks, such as supervised training of a classifier. Note that in the pretraining phase, the encoder $\mathcal{F}$ is trained in a self-supervised fashion without access to any labels. We base our method on the work of Eldele et al. (2021) and adapt their method *TS-TCC* to enable the training on multiple datasets. Similar to their work, we use a simple 1D convolutional model with three layers as the encoder $\mathcal{F}$. As shown in Figure 2, we merge a pair of time series through interpolation and then derive two augmented variants to calculate two distinct pretraining losses, namely the **T**emporal **C**ontrastive (TC) and **S**oft **I**nterpolation **C**ontextual **C**ontrastive (SICC) loss. Eventually, they are combined into a single training objective $\mathcal{L}_{\text{Total}}$ and are optimized jointly, yielding us the XIT architecture. The entire procedure is listed in the Appendix at Algorithm 1.

### 2.1 XD-MIXUP AND DATA AUGMENTATION

We now construct a pretraining task to train the joint encoder $\mathcal{F}$. Given two different time series $x_i$ and $x_j$, we first generate a pointwise interpolated variant $\widetilde{x} \in \mathbb{R}^T$ by

$$\widetilde{x}_i = \lambda_i x_i + (1 - \lambda_i)x_j, \quad \lambda_i \sim \text{Beta}(\alpha, \alpha). \tag{1}$$

Here, $\alpha > 0$ is the shape parameter of the symmetric Beta distribution. Since $\lambda_i \in [0, 1]$, $\widetilde{x}_i$ is a convex combination of $x_i$ and $x_j$. This is inspired by the label smoothing method MixUp (Zhang et al., 2022), which is known to improve robustness to outliers, training stability, and calibration (Thulasidasan et al., 2019) in the image domain and has already been applied to time series by Wickstrøm et al. (2022). Besides the usual benefits of data augmentation, we use it to generate data points

---

[1]pronounced «exit»

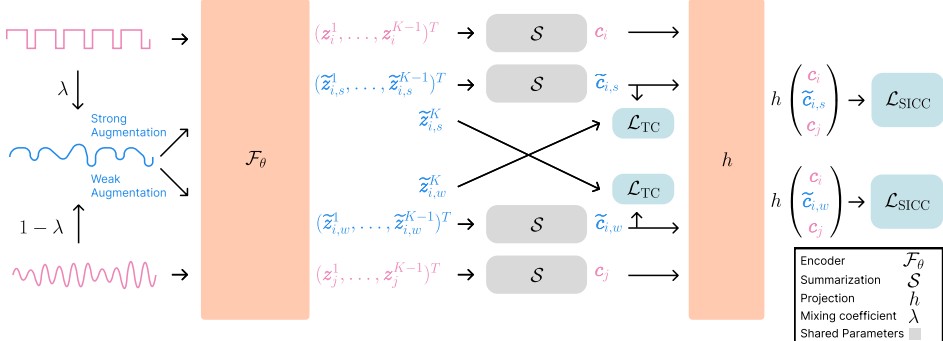

Figure 2: Our proposed XIT architecture. From two time series of a mini-batch, we generate a randomly interpolated variant that gets augmented twice and projected several times along with the original time series. Eventually, we compute two losses to define the overall pretraining objective.

between the different clusters of time series induced by training on multiple datasets, typically each consisting of one or many sub-clusters. This is especially relevant since the eventual downstream task might consist of some time series not seen in the pre-training phase and, therefore, might lie outside of the clusters.

A mini-batch consists of randomly sampled time series from each of the included datasets. We select the pairs for interpolation from a single mini-batch on the fly while optimizing. Since a batch of size $B$ does not contain duplicate time series, we can build distinct pairs by interpolating between consecutive pairs of time series, like $(\boldsymbol{x}_1, \boldsymbol{x}_2), (\boldsymbol{x}_2, \boldsymbol{x}_3), \ldots, (\boldsymbol{x}_B, \boldsymbol{x}_1)$ with independently sampled $\lambda_i$. We are then left with a batch of $B$ interpolated time series $\widetilde{\boldsymbol{x}}_i$, where $i \in \{1, \ldots, B\}$, in addition to the original time series. We deliberately do not consider *all* possible combinations within a mini-batch to not dramatically increase the batch size, which would incur high computational costs in the pretraining loss computations. Following TS-TCC, we then apply a strong and a weak augmentation to the time series $\widetilde{\boldsymbol{x}}_i$ to obtain $\widetilde{\boldsymbol{x}}_{i,s}$ and $\widetilde{\boldsymbol{x}}_{i,w}$.

## 2.2 PRETRAINING LOSS

The strongly and weakly augmented samples are then used to compute the loss $\mathcal{L}_{\text{Total}}$, which consists of a weighted combination of the temporal contrastive loss $\mathcal{L}_{\text{TC}}$ and the soft interpolation contextual contrastive loss $\mathcal{L}_{\text{SICC}}$:

$$\mathcal{L}_{\text{Total}} = \beta \mathcal{L}_{\text{TC}} + (1 - \beta)\mathcal{L}_{\text{SICC}}. \tag{2}$$

The weight $\beta \in [0, 1]$ is determined by hyperparameter search. The TC loss guarantees that a time series representation captures its unique, relevant features and differentiates from others. The SICC loss in combination with MixUp ensures meaningful connections between the various datasets seen during pretraining by enforcing a well-structured latent space between them.

As shown in Figure 2, the next step after XD-MixUp is to encode each time series $\widetilde{\boldsymbol{x}}_{i,s}$ into a sequence of $K$ embedding vectors $\left(\boldsymbol{z}_{i,s}^1, \ldots, \boldsymbol{z}_{i,s}^K\right) = \boldsymbol{z}_{i,s} = \mathcal{F}(\boldsymbol{x}_{i,s}, \boldsymbol{\theta})$ using the convolutional encoder $\mathcal{F}$, where $\boldsymbol{z}_{i,s}^k \in \mathbb{R}^Z$. The time series $\widetilde{\boldsymbol{x}}_{i,w}$ and the original $\boldsymbol{x}_i$ are encoded into $\widetilde{\boldsymbol{z}}_{i,w}$ and $\boldsymbol{z}_i$, respectively. We then use a shared summarization model $\mathcal{S}(\cdot)$ to condense the first $K - 1$ embedding vectors $\boldsymbol{z}^{1:(K-1)}$ into a single context $\boldsymbol{c} \in \mathbb{R}^C$, for all time series individually. Working with these summary contexts instead of with the individual embedding vectors greatly simplifies the computation of the pretraining losses and makes it more efficient. In addition, it promotes learning more high-level features that can compress all embedding vectors into smaller representations (Eldele et al., 2021). We will now provide an in-depth explanation of both TC and SICC loss.

### 2.2.1 TEMPORAL CONTRASTING

The TC loss (Eldele et al., 2021) is computed by solving a cross-forecasting task. The context derived from the weakly augmented embedding $\widetilde{\boldsymbol{c}}_w$ is used to predict the last embedding vector $\widetilde{\boldsymbol{z}}_s^K$ of the strong embedding, and vice versa. This is done using a jointly learned similarity measure

$g(\widetilde{c}, \widetilde{z}^K) = \exp(\widetilde{c}^T W \widetilde{z}^K)$. Here, $W \in \mathbb{R}^{C \times Z}$ is a matrix that aligns the dimensions of the two vectors. The task is to maximize the similarity to the differently augmented time series embedding of the same time series while minimizing the similarity to the other embeddings from the mini-batch. This favors representations that are invariant to the augmentations being applied. Overall, we compute the loss as follows:

$$\mathcal{L}_{\text{TC}}^s = -\frac{1}{B} \sum_{i=1}^{B} \log \left( \frac{g(\widetilde{c}_{i,w}, \widetilde{z}_{i,s}^K)}{\sum_{j=1}^{B} g(\widetilde{c}_{i,w}, \widetilde{z}_{j,s}^K)} \right) \qquad \mathcal{L}_{\text{TC}}^w = -\frac{1}{B} \sum_{i=1}^{B} \log \left( \frac{g(\widetilde{c}_{i,s}, \widetilde{z}_{i,w}^K)}{\sum_{j=1}^{B} g(\widetilde{c}_{i,s}, \widetilde{z}_{j,w}^K)} \right)$$

We then average them to obtain $\mathcal{L}_{\text{TC}} = \frac{1}{2} (\mathcal{L}_{\text{TC}}^s + \mathcal{L}_{\text{TC}}^w)$. The parts $\mathcal{L}_{\text{TC}}^s$ and $\mathcal{L}_{\text{TC}}^w$ are called **n**ormalized **t**emperature-scaled **cross-ent**ropy loss (NT-Xent) (Chen et al., 2020). That loss is also called InfoNCE (from **N**oise **C**ontrastive **E**stimation) since van den Oord et al. (2018) have shown that it optimizes a lower bound on the mutual information $I(\widetilde{c}; \widetilde{z})$ between the context vectors $\widetilde{c}$ and their corresponding embeddings $\widetilde{z}$. Larger batch sizes $B$ yield tighter bounds.

### 2.2.2 SOFT INTERPOLATION CONTEXTUAL CONTRASTING

Our novel SICC loss aligns the information in the augmented time series context vector to the non-augmented contexts. This enforces the encoder $\mathcal{F}$ to be invariant to the selected augmentations, thereby capturing higher-level concepts. We want the context to contain the information of the source time series pair $(x_i, x_j)$ used to form the interpolated time series $\widetilde{x}_i$ depending on the interpolation coefficient $\lambda_i$. Namely, if $\lambda \approx 0$ then $(\widetilde{x}_i, x_i)$ should be a positive pair and $(\widetilde{x}_i, x_j)$ a negative pair. If $\lambda \approx 1$, then the positive/negative relations switch. We, therefore, extend the normal notion of *hard* positive-negative examples to a *soft* variant that treats pairing within the interpolated time series group proportional to $\lambda$ and still considers the rest of the mini-batch to be hard negative samples. Our approach thereby differs from the loss of Sohn (2016) and Chen et al. (2020) used in TS-TCC, where the contextual alignment is solely performed between the two augmented time series.

To enforce the entire embeddings $z$ to be aligned, we directly use the contexts $c$ that we computed with the summarization model $\mathcal{S}$. Since we want the information content of a positive pair to match but do not require the concrete representation to be exactly the same, we further project the contexts $c$ using a two-layer learned MLP, obtaining $\kappa = h(c) = \text{Linear}(\text{ReLU}(\text{BatchNorm1D}(\text{Linear}(c))))$. Here, $\text{Linear}(\xi) = W\xi + b$ such that the resulting vector has half the dimension of the input vector, i.e., that $\kappa \in \mathbb{R}^{C/4}$.

Let $(\kappa_{i,l}, \kappa_{i,s}, \kappa_{i,r})$ with $i \in \{1, \ldots, B\}$ be the mini-batch of $B$ triples consisting of either the strongly or weakly augmented time series projection $\kappa_i^s$ or $\kappa_i^w$ along with the projection of the left $\kappa_{i,l}$ and right time series $\kappa_{i,r}$ that were interpolated between with $\lambda_i$ to form the augmented one in eq. (1). We arrange these into two sets of vectors of length $3B$:

$$\mathfrak{B}^s = (\kappa_{1,l}, \ldots, \kappa_{B,l}, \quad \kappa_{1,s}, \ldots, \kappa_{B,s}, \quad \kappa_{1,r}, \ldots, \kappa_{B,r}),$$
$$\mathfrak{B}^w = (\kappa_{1,l}, \ldots, \kappa_{B,l}, \quad \kappa_{1,w}, \ldots, \kappa_{B,w}, \quad \kappa_{1,r}, \ldots, \kappa_{B,r}).$$

The loss function is then computed as the average $\mathcal{L}_{\text{SICC}} = \frac{1}{2} (\mathcal{L}_{\text{SICC}}(\mathfrak{B}^s) + \mathcal{L}_{\text{SICC}}(\mathfrak{B}^w))$, where the individual parts are defined as:

$$\mathcal{L}_{\text{SICC}}(\mathfrak{B}) = \frac{1}{B} \sum_{i=1}^{B} \Big[ \ell(\mathfrak{B}, i, B+i, 1-\lambda_i) + \ell(\mathfrak{B}, B+i, i, 1-\lambda_i) + \tag{3}$$
$$\ell(\mathfrak{B}, B+i, 2B+i, \lambda_i) + \ell(\mathfrak{B}, 2B+i, B+i, \lambda_i) \Big]$$

$$\ell(\mathfrak{B}, i, j, \mu) = -\log \left( \frac{\exp(\mu \sin(\mathfrak{B}_i, \mathfrak{B}_j)/\tau)}{\sum_{k=1}^{3B} \mathbb{1}_{k \neq i} \exp(\sin(\mathfrak{B}_i, \mathfrak{B}_k)/\tau)} \right) \tag{4}$$

Here, $\text{sim}(x, x') = \frac{x^T x'}{\|x\|_2 \|x'\|_2}$ denotes the cosine similarity, $\mathbb{1}$ is an indicator variable that is 1 if $k \neq i$ and 0 otherwise, and $\tau$ is the temperature parameter. In Equation (1), $\lambda_i$ scaled the *distance* of $\widetilde{x}_i$ to $x_i$ and $1 - \lambda$ the distance to the other time series, $x_j$. Since we now want to scale *similarities* proportionally, we have to reverse the roles of $\lambda$ and $1 - \lambda$. In eq. (4), since we minimize the negative of the fraction, we optimize for maximizing the numerator (i.e., the similarity of the positive pair $(i, j)$) and minimizing the denominator (i.e., the similarity of all other negative pairs $(i, k)$). The computation of $\ell$ can be numerically stabilized using the log-sum-exp trick.

We have now defined the three core components of our approach and how they connect. First, we interpolate between datasets with XD-MixUp, to then apply the TC and SICC losses. Together, they form the complete XIT pretraining procedure, whose efficacy we can now asses in the next section.

## 3 EXPERIMENTS

After having laid out the motivation and formal foundation of XIT, we will answer these key research questions with our experiments: **(Q1)** Do multiple datasets help to learn a more general representation that is easier to transfer to seen and unseen datasets? **(Q2)** Is an encoder pretrained in a self-supervised fashion more helpful than directly learning a classifier, especially in low-data scenarios? **(Q3)** Which key components of our proposed XIT procedure cause the improvements that we observe? **(Q4)** How discriminative are the learned representations regarding inter- and intra-dataset structures? The remainder of this section will provide an overview of our experimental setup, with further details provided in Appendix A.2. Subsequently, we will present the results and discoveries.

**Datasets**  We compare our method with the baselines on five diverse univariate classification datasets inspired by the TF-C *N-to-one* setting (Zhang et al., 2022, Appendix K) whose setup we follow. The Appendix gives an overview of the datasets and their characteristics in Table 4. We also evaluate on the large UCR Time Series Classification repository. (1) *Sleep-EDF* (sometimes called Sleep-EEG) (Kemp et al., 2000) obtained from the PhysioNet repository (Goldberger et al., 2000) is a dataset of polysomnographic recordings of whole-night sleep cycles in European Data Format (EDF). We followed the common practice of using the Fpz-Cz channel of the EEG signal to classify the five sleep stages. (2) The *FD-A* dataset (Lessmeier et al., 2016) consists of measurements in a real-world industrial setting. The measured current of an electric motor is used to distinguish three conditions of an attached ball bearing. (3) In the *HAR* dataset (Anguita et al., 2013), the wrist movements of 30 subjects are recorded while they perform activities of daily living. We use the acceleration along the x-axis of the nine available sensors to identify one of six activities. (4) The *ECG* dataset (Clifford et al., 2017) stems from the 2017 PhysioNet Challenge (Goldberger et al., 2000) and contains single-lead ECG recordings with four different types of cardiac arrhythmias lasting from 9 to 60 seconds, split into 5-second windows. (5) *Epilepsy* (Andrzejak et al., 2001) contains single-channel EEG measurements, with the binary classification task of determining if the patient has a seizure. (6) Finally, the *UCR Time Series Classification* repository (Dau et al., 2019) contains 128 datasets of a wide range of domains, time series lengths, sample counts, and numbers of classes. We used all datasets of up to 600 time steps, resulting in 100 datasets as shown in Table 5 in the Appendix. We did not include longer ones to limit the discrepancy with shorter datasets, which would contain increasingly more padding.

**Data preparation and augmentation**  For a level comparison, we ensure that all time series datasets are equal in length. For Epilepsy, we dropped the second half of the series, while for all other datasets, we front-padded with zeros to obtain a total of 1,500 time steps in each series. We use the same augmentations as in the supplementary TS-TCC implementation: magnitude scaling as weak augmentation and *permutation-and-jitter* as strong augmentation, with the same parameters as TS-TCC wherever available for the datasets.

**Baselines**  In total we compare XIT against multiple baselines to evaluate its effectiveness. To this end, we employed the following five self-supervised pretraining methods: (1) TS-TCC (Eldele et al., 2021), which uses temporal and contextual contrasting between weakly and strongly augmented time series to learn an embedding. (2) TF-C (Zhang et al., 2022), which learns and aligns a time and frequency domain encoding into a joint representation, again with contrastive methods. (3) TS2Vec (Yue et al., 2022), which performs masking and contrasting hierarchically. (4) TNC (Tonekaboni et al., 2020) aligns neighboring windows with their respective latent representation. (5) T-Loss (Franceschi et al., 2019) utilizes time-based negative sampling in conjunction with a triplet loss to learn an encoding. Additionally, the *supervised* model performs no pretraining, i.e., it uses a randomly initialized encoder that is then used for finetuning.

**Evaluation**  We evaluate the classification mainly with AUROC and macro-averaged F1 scores since the accuracy metric is unreliable in the face of uneven class distributions. We still provide it for future reference. We also employed statistical ranking tests and diagrams to evaluate the significance of the observed critical difference (CD) in classifier performance (Dau et al., 2019).

Table 1: The results of pretraining on an increasing number of pretraining datasets «PT» and individually finetuning on all five targets for our model and the baselines. Finteuning was limited to 2.5% of the data. The first row shows the supervised baseline. The values indicate the mean AUROC score in percent and its standard deviation over five random seeds. **Bold** denotes the best overall, and underlined is the best within each number of pretraining datasets. Higher is better.

| PT | Model | Sleep-EDF | FD-A | HAR | ECG | Epilepsy |
|---|---|---|---|---|---|---|
| | Superv. | 76.47 ±2.4 | 86.39 ±2.1 | 85.80 ±0.6 | 46.34 ±0.5 | **98.19 ±0.3** |
| 1 | XIT | 85.57 ±0.9 | **96.52 ±0.5** | **89.78 ±0.7** | 41.44 ±0.8 | 97.45 ±0.7 |
| | TS-TCC | 87.55 ±0.6 | 66.64 ±2.7 | 52.92 ±8.2 | 40.67 ±0.6 | 93.03 ±8.3 |
| | TF-C | 73.22 ±2.5 | 93.19 ±2.8 | 83.01 ±0.5 | 39.37 ±1.7 | 47.64 ±24.2 |
| 2 | XIT | 86.17 ±1.0 | 95.85 ±0.5 | 88.50 ±1.0 | 44.22 ±0.4 | 97.69 ±0.6 |
| | TS-TCC | **87.72 ±0.6** | 36.99 ±2.3 | 52.50 ±11.9 | 41.25 ±1.3 | 95.28 ±2.3 |
| | TF-C | 67.13 ±6.1 | 88.80 ±2.9 | 84.44 ±1.5 | 38.54 ±1.2 | 52.27 ±16.0 |
| 3 | XIT | 85.74 ±1.0 | 95.75 ±0.3 | 88.54 ±0.6 | 46.43 ±0.9 | 97.87 ±0.5 |
| | TS-TCC | 86.38 ±1.2 | 31.76 ±3.7 | 71.53 ±8.2 | 41.69 ±0.7 | 90.20 ±6.3 |
| | TF-C | 72.92 ±3.8 | 88.74 ±1.2 | 82.43 ±3.2 | 41.35 ±1.5 | 64.19 ±15.0 |
| 4 | XIT | 84.39 ±0.4 | 95.79 ±0.7 | 88.11 ±0.3 | **47.30 ±1.1** | 97.50 ±0.3 |
| | TS-TCC | 82.52 ±1.4 | 61.22 ±1.2 | 81.32 ±0.5 | 42.31 ±1.0 | 96.57 ±0.7 |
| | TF-C | 63.37 ±3.4 | 81.59 ±1.8 | 82.72 ±0.8 | 44.89 ±2.0 | 66.93 ±23.4 |

Figure 3: CD diagram for the *many2many* scenario (see Table 1) when pretraining on four datasets and five seeds. The classifiers were ranked by AUROC score, with higher average ranks being further to the right. A connecting line indicates that there is no significant difference in performance according to the Friedman test with $\alpha = 0.01$.

## 3.1 SELF-SUPERVISED PRETRAINING ON MULTIPLE DATASETS: (Q1) & (Q2)

To determine whether we can learn a single encoder on multiple datasets, we first reproduce the experimental *many2many* setup of TF-C. Hereby, we pretrain on the first 1, 2, 3, and 4 datasets of Sleep-EDF, FD-A, HAR, and ECG and finetune on those and additionally on Epilepsy for a more complete overview. The results for finetuning on 2.5% of the datasets are shown in Table 1. Due to space constraints, it only includes AUROC scores, with Accuracy and Macro F1 scores deferred to Tables 7 and 8 in the Appendix. The results for scaling to larger fractions of the finetuning datasets (5%, 10%, 50%, and 100%) can be found in Appendix A.3. This experiment illustrates that TF-C exhibits unstable training, leading to unpredictable performance with each new dataset. In contrast, XIT demonstrates robustness, with minimal decreases and often significant increases across all cases. TS-TCC performs adequately with five datasets but lacks robustness, exhibiting similar instability to TF-C when more data is leveraged. Notably, in datasets like ECG, consistent increases are observed across all models, yet XIT benefits the most. For Epilepsy, supervised is the most effective approach, although XIT is nearly as accurate, coming to within 0.5 percentage points of the supervised approach. We further apply CD tests and determine that the effect is significant with high confidence, as demonstrated in Figure 3. These experiments show that—in contrast to the findings of Zhang et al. (2022)—pretraining on more than one dataset is indeed feasible and beneficial when compared to supervised.

To investigate whether scaling to much larger numbers of datasets can again increase the finetuning performance, we continue to apply the methodology to large portions of the UCR repository. Here, 100 datasets have a fixed sequence length of up to 600. We pretrained on up to 75 different datasets, where we subsequently finetuned both on all 100 datasets (Figure 4a) and 25 of the held-out ones (Figure 4b). We sample each domain with the same frequency (see Table 5 in the Appendix). Within each domain,

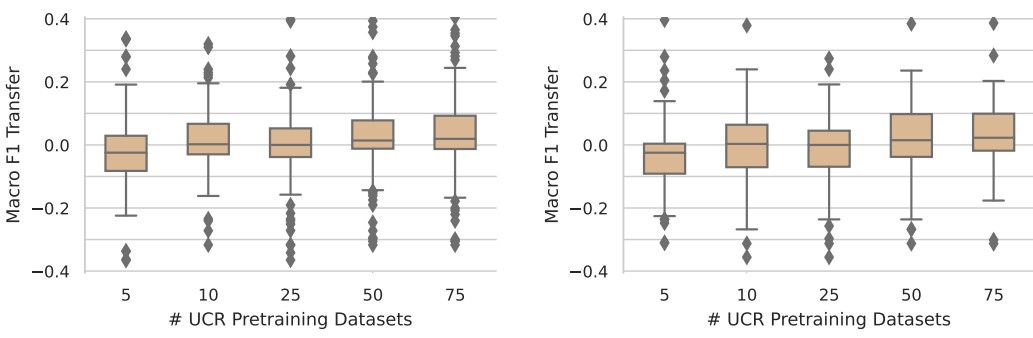

(a) Finetuning on the 100 UCR datasets.  (b) Finetuning on a hold-out set of 25 datasets each.

Figure 4: This box plot shows the transfer surplus over supervised training measured in Macro F1 score difference after pretraining XIT on increasingly large subsets of the UCR repository with three folds each. Higher is better.

we sampled time series data points for training according to the size of the dataset. We create three folds of each dataset selection, where we each perform the pretraining and finetuning steps with independently initialized models. We first observe that in Figure 4a, increasing the number of datasets in the pretraining phase increases the benefit of pretraining over supervised training since the Macro F1 transfer becomes increasingly positive. The effect is even more pronounced when only transferring to entirely unseen datasets, where the increased diversity aids in learning more general representations. In the case of pretraining on 75 datasets, XIT outperforms direct supervised training in 64.7% of all datasets, while in the hold-out evaluation, this is still true for an impressive 62.5% datasets. In conclusion, XIT effectively learns time series representations from multiple, diverse datasets.

To further position XIT among existing approaches, we compare it to five baselines. We evaluate on challenging 25 UCR datasets not seen during pretraining on the other 75 ones. Table 2 shows the competitive and state-of-the-art performance of XIT. For ease of comparison, we follow Demšar (2006) and append ranks for each method. In general, we outperform the purely constrastive methods. Even when compared to the mixed reconstructive-contrastive method TS2Vec, we outperform it overall.

## 3.2 ABLATION STUDIES: (Q3)

We conduct a series of ablation experiments to determine if the observed effects were due to the key components of XIT. We pretrained on the same three folds of 75 UCR datasets and finetuned on all 100 datasets. We ablated by systematically omitting our method's three key components XD-MixUp, TC loss, and SICC loss. Note that SICC is inherently linked to the presence of XD-Mixup, so SICC in isolation is not feasible. Similarly, only XD-MixUp without a loss is not a valid training procedure. The results are presented in Table 3. Our method XIT is superior to any ablated models when looking at the ranks for accuracy, AUROC, and Macro F1 scores. In particular, the high AUROC score means we have a high overall probability of correctly classifying classes, whereas the high Macro F1 score shows we can also correctly classify underrepresented classes. Therefore, we conclude that all three building blocks of our combined pretraining procedure XIT are relevant for the performance increases observed in the previous section.

## 3.3 INSPECTING THE LEARNED REPRESENTATIONS: (Q4)

To gain insight into the effects of our proposed method, we examined the structure of multiple embedded time series datasets. When embedded with a newly initialized encoder, we achieve a Davies-Bouldin Index (DBI) (Davies & Bouldin, 1979) of 7.63. After pretraining on 75 datasets, this score improved to 7.32. This relatively small decrease suggests that our method does not need to drastically alter the overall structure, but rather rearranges the intra-dataset structure. This effect can be seen qualitatively in the appendix in Figure 6, which shows a projection of the latent representation learned by XIT of multiple unseen datasets. We observe a major structuring effect without any finetuning or the introduction of any labels. This demonstrates that in the majority of cases, the

Table 2: This table shows the classification performance of XIT in comparison to baselines when pretrained on 75 datasets of the UCR repository. The shown Macro F1 scores were computed on three independent seeds and 25 datasets not seen during pretraining (similar to the hold-out in Figure 4b).

| Target Dataset | XIT | TS-TCC | TF-C | TS2Vec | TNC | T-Loss |
|---|---|---|---|---|---|---|
| ArrowHead | 50.5±9.2 | 47.5±2.1 | 36.9±11.9 | **57.4±0.4** | 36.6±8.1 | 29.5±2.2 |
| BeetleFly | **77.6±6.4** | 66.5±11.0 | 46.3±7.6 | 68.0±5.9 | 67.2±14.5 | 61.7±6.9 |
| BirdChicken | 66.6±12.6 | 63.0±5.8 | 56.4±20.0 | **88.0±8.1** | 57.8±8.2 | 61.5±7.7 |
| Car | 47.7±2.1 | 60.2±10.6 | 25.4±6.0 | **73.0±2.2** | 29.9±14.1 | 23.5±6.0 |
| Crop | **53.7±0.7** | **53.7±1.2** | 47.1±0.8 | 40.4±2.5 | 19.6±9.8 | 14.1±1.1 |
| Fish | **59.4±1.9** | 55.0±7.9 | 31.8±6.9 | 44.6±7.3 | 12.5±6.3 | 08.5±4.6 |
| FreezerRegularTrain | **77.2±0.3** | 76.2±0.3 | 76.9±1.2 | 76.9±0.5 | 73.1±8.0 | 58.3±3.0 |
| GunPoint | 72.1±6.3 | 78.9±2.1 | 58.7±17.0 | **94.4±2.8** | 62.4±22.4 | 64.6±12.0 |
| GunPointMaleV.Female | 92.7±2.2 | **95.7±3.7** | 77.6±5.7 | 86.7±3.4 | 75.5±8.0 | 74.3±3.8 |
| Lightning7 | 65.9±1.2 | 50.5±3.1 | 17.5±3.6 | **78.8±4.7** | 35.6±17.2 | 37.3±2.7 |
| MedicalImages | **22.2±1.0** | 06.8±0.0 | 12.2±3.0 | 06.8±0.0 | 09.7±4.3 | 08.2±2.5 |
| MiddlePh.O.AgeGroup | 31.8±4.7 | 20.2±7.3 | **39.6±1.8** | 21.1±2.3 | 23.9±13.1 | 25.4±14.9 |
| MiddlePh.O.Correct | **36.3±0.0** | **36.3±0.0** | **36.3±0.0** | **36.3±0.0** | **36.3±0.0** | **36.3±0.0** |
| MiddlePhalanxTW | 19.9±3.7 | 14.3±2.3 | **24.3±2.0** | 15.6±0.7 | 14.3±7.4 | 14.6±8.1 |
| OSULeaf | 41.9±0.6 | 41.7±3.0 | 34.1±2.8 | **46.1±0.6** | 14.6±6.6 | 12.3±2.0 |
| ProximalPh.O.Correct | 41.0±0.7 | 41.0±0.7 | **46.8±6.2** | 40.6±0.0 | 41.4±1.8 | 40.6±0.0 |
| ShapesAll | **70.9±0.2** | 66.4±0.8 | 28.4±6.9 | 50.1±3.9 | 08.0±5.5 | 03.0±0.9 |
| SonyAIBORobotSur.1 | 30.3±0.2 | 30.0±0.0 | **76.0±10.3** | 38.1±4.7 | 47.8±13.1 | 45.7±14.0 |
| SonyAIBORobotSur.2 | 51.6±11.8 | 50.9±11.6 | **86.2±1.1** | 70.4±4.2 | 63.7±5.7 | 65.4±1.3 |
| Symbols | 72.0±0.5 | 74.7±1.3 | 36.7±3.0 | **90.7±1.9** | 45.1±19.5 | 53.6±16.7 |
| Tiselac | **29.3±0.3** | 26.3±0.7 | 21.2±TBA | 18.5±1.2 | 15.3±2.7 | 12.3±0.2 |
| ToeSegmentation2 | 69.6±7.4 | 74.3±9.6 | 46.9±18.4 | **76.0±3.0** | 56.3±8.4 | 57.8±0.3 |
| UWaveGestureLib.X | **77.1±0.3** | 69.7±0.5 | 55.1±5.7 | 64.1±0.2 | 32.3±10.6 | 21.1±2.6 |
| UWaveGestureLib.Z | **68.3±1.3** | 61.3±1.9 | 51.4±3.2 | 55.5±1.1 | 27.1±11.8 | 22.9±3.1 |
| WordSynonyms | **35.1±4.4** | 17.3±0.6 | 06.6±1.1 | 03.3±0.5 | 04.0±1.3 | 03.1±1.7 |
| Average Macro F1 ↑ | **54.4±3.2** | 51.1±3.5 | 43.1±6.1 | 53.7±2.5 | 36.4±9.1 | 34.2±4.7 |
| Rank ↓ | **2.10±1.2** | 3.06±1.6 | 3.56±1.8 | 2.82±1.6 | 4.48±1.0 | 4.98±1.1 |

Table 3: The ablation studies we performed to answer (Q4). We followed the same evaluation as in Table 2. Ranks closer to one are better, **bold** denotes best. The results demonstrate that each of the key components, XD-MixUp, TC loss, and SICC loss, contribute to the overall performance of XIT.

| Pretraining Component | AUROC rank ↓ | Accuracy rank ↓ | Macro F1 rank ↓ |
|---|---|---|---|
| XD-MixUp + SICC + TC (XIT) | **1.800 ±0.91** | **1.780 ±0.89** | **1.780 ±0.84** |
| XD-MixUp + SICC | 3.560 ±0.87 | 3.700 ±0.56 | 3.580 ±0.79 |
| XD-MixUp + TC | 2.060 ±0.94 | 1.920 ±0.84 | 1.980 ±0.91 |
| TC | 2.580 ±0.91 | 2.600 ±0.78 | 2.660 ±0.81 |

pretraining losses induce beneficial structure in the latent space, which is consistent with the results of Figure 4.

## 4 RELATED WORK

Pretraining is a key element of current deep learning, allowing state-of-the-art outcomes in areas with scarce labels or data by utilizing a shared representation as the basis for adapting to the target domain. Although it has been extensively studied for domains such as natural language processing and computer vision, it still remains a challenge for the time series domain (Ma et al., 2023). We can generally differentiate between supervised, unsupervised, and self-supervised pretraining. The core idea for the former is to utilize labels to steer the representation learning, while the latter two

approaches work without any labels. Due to the wide availability of unlabeled time series datasets, we will focus on self-supervised methods in this paper.

**Self-Supervised Time Series Pretraining**    Several methods have been proposed to pretrain models on unlabeled datasets. The three main types of losses to optimize are reconstruction, pseudo-labels, and contrastive methods (Ma et al., 2023). The SimMTM framework (Dong et al., 2023) reconstructs time series from multiple masked variants and series-wise similarities within and across domains. Further models based on reconstructions are Ti-MAE Li et al. (2023) and its extension TimeMAE Cheng et al. (2023). Contrastive methods, on the other hand, need means to generate view pairs, e.g., as proposed in LEAVES (Yu et al., 2022), by Tang et al. (2020), or in PAITS (Beebe-Wang et al., 2023). Shi et al. (2021) learn long-term dependencies using Dynamic Time Warping (DTW) (Sakoe & Chiba, 1978). Kiyasseh et al. (2021) specifically apply their contrastive learning method CLOCS to ECG signals. In TS2Vec by Yue et al. (2022), instance-wise and temporal hierarchical contrasting is used to capture multiscale contextual information and dynamics. TS-TCC (Eldele et al., 2021) is a pretraining framework involving two views generated by weak and strong augmentations, which we also use in XIT. This is then fed into a temporal and contextual contrasting module using the NTXent loss (Sohn, 2016) for learning robust and discriminative representations. Furthermore, the authors indicate that it is effective in transfer learning scenarios with few-labeled targets. To use spectral information in contrasting, Zhang et al. (2022) developed TF-C, allowing the model to align the time and frequency domains with the respective views. Wickstrøm et al. (2022) propose MixUp (Zhang et al., 2018) for the time domain, where two samples are combined by a sampled parameter $\lambda$, which is predicted as a pseudo-label. Furthermore, the two mixed views are aligned via the same contrastive NTXent loss used in TS2Vec, TS-TCC, TF-C, and our method XIT. Tonekaboni et al. (2020) propose TNC, utilizing time series windows where ones with close-proximity share similar latent representations. In addition, Franceschi et al. (2019) propose T-Loss, which employs time-based negative sampling along with a triplet loss to learn an encoding.

**Multi-Dataset Pretraining**    While it is common to pretrain on a single source dataset, there is very little research in the area of multi-dataset pretraining, especially in the context of time series classification (Ma et al., 2023). This arises from the fact that applying multiple datasets in a suboptimal setting may drastically decrease the performance (Zhang et al., 2022), leading to a so-called *negative transfer*. Other works (Gikunda & Jouandeau, 2021; Tseng et al., 2023; Brüsch et al., 2023) leverage multiple datasets in homogeneous settings where source and target distributions match. As far as the authors are aware, Kashiparekh et al. (2019) and Zhang et al. (2022) are the only works that investigate proper multi-dataset pretraining. However, the former applies it in a supervised and very inflexible way, using one encoding head per source dataset, and the latter reports significant challenges when applying their method TF-C to multiple datasets at once, which they call *many-to-one* setting. They note a clear drop in performance when increasing the number of datasets from one to two, three, and four.

## 5 CONCLUSION & FUTURE WORK

Our research presents a paradigm shift in time series pretraining. Contrary to prevailing beliefs, our findings illustrate the possibility and effectiveness of multi-dataset pretraining for time series. By introducing XIT, consisting of XD-MixUp along with the SICC and TC losses, we have carved a promising path in self-supervised contrastive pretraining for time series. Our empirical evaluations showcased the efficacy of this method, especially in low-data regimes, against both supervised training and other self-supervised pretraining techniques. In essence, not only have we debunked the myth that multi-dataset pretraining is infeasible for time series, but we have also opened the door for further advancements in leveraging multiple datasets—beyond simultaneously using 75 datasets.

While our study has advanced time series pretraining, several promising directions beckon further exploration. The versatility of our approach needs evaluation on further tasks like forecasting and anomaly detection. We are eager to explore model reprogramming (Yang et al., 2021) to enhance adaptability and further decrease negative transfer. While we utilized MixUp augmentation, exploring specialized interpolations like DTW may yield further insights. Furthermore, we want to explore the potential of our SICC loss and integrated interpolation mechanism in other time series models and different modalities. Much like in NLP, future work might consider creating compound datasets with special attention to the types and proportions of contained data.

REPRODUCIBILITY

We acknowledge the significance of reproducibility in scientific research and have taken multiple steps to ensure the strength and replicability of our work.

- **Code:** Our implementation is accessible on GitHub at `https://anonymous.4open.science/r/TS-XIT`. We have used publicly available software and libraries to guarantee accessibility and have comprehensively described the architecture, software, versions, and hyperparameters in the Appendix A.2. Our code is deterministic, incorporating seeds for all random number generators to guarantee the replicability of results. We attempted to include most of the code used to create the result tables and figures in this manuscript.

- **Datasets:** This study only utilizes publicly available datasets that have been correctly cited. Furthermore, the authors contribute to an open-source repository containing all the datasets used in this work, which will be made available upon acceptance.

- **Architecture and Algorithm Details:** We have provided thorough descriptions and formulations of our architecture in the main text, supplemented by additional clarifications and implementation details in the Appendix A.2, ensuring a clear understanding of our contributions and facilitating reproduction. This documentation is intended to provide researchers with all the necessary information to replicate our experiments accurately.

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

# A  APPENDIX

## A.1  OUR PROPOSED PROCEDURE

The complete XIT procedure to perform both pretraining and subsequent finetuning is given in Algorithm 1.

## A.2  EXPERIMENTAL DETAILS

This section gives more details for the experimental evaluation that should aid in reproducing our results. First of all, Table 4 lists the datasets we used in the many2many evaluation. Similarly, Table 5 compares the datasets in the UCR repository, which are freely available from `https://timeseriesclassification.com`.

**Implementation details**  We used *PyTorch* (Paszke et al., 2019) version 2.0 as the base framework for all models. We performed almost all training in 16-bit mixed precision to save resources. For TS2Vec, we stayed very close to the reference implementation and therefore trained in 32-bit precision, used the existing hyperparameters for pretraining (see Table 6), and no early stopping in finetuning. We used early stopping after four training epochs without improvements in the AUROC score for all other finetuning/supervised experiments. We ensured that we trained for at least 40 steps before stopping and up to a maximum of 2000 steps. The Adam optimizer with $\beta_1 = 0.9$ and $\beta_2 = 0.999$ was used to optimize all models with the hyperparameters given in Table 6. For the special case of the disproportionally large *Tiselac* dataset, we increased the batch size to 256 for faster completion. For the data interpolation in eq. (1), we set $\alpha = 0.2$ as determined by a hyperparameter search. In eq. (2) we set $\beta = 0.25$ according to our hyperparameter search, effectively giving three times more weight to $\mathcal{L}_{\text{SICC}}$ than to $\mathcal{L}_{\text{TC}}$. In $h(c)$, $\text{BatchNorm1D}$ normalizes the vectors per dimension by subtracting the mean and dividing by the empirical standard deviation. We used the default PyTorch configuration, maintaining a running average of the elements within the mini-batches. We set the temperature $\tau$ in eq. (4) to 0.2. We based our implementation of the baselines on the official repositories of TS-TCC (`https://github.com/emadeldeen24/TS-TCC`), TF-C (`https://github.com/mims-harvard/tfc-pretraining`), TNC `https://github.com/sanatonek/TNC_representation_learning`, T-Loss `https://github.com/White-Link/UnsupervisedScalableRepresentationLearningTimeSeries` and TS2Vec (`https://github.com/yuezhihan/ts2vec`), respectively.

**Encoder & Summarization Model**  We used a simple and efficient encoder $\mathcal{F}$ with three residual convolution layers and configured it as in the work of Wang et al. (2017) and TS-TCC. For the model $\mathcal{S}$ used to obtain the context vectors $c$, we used the exact same transformer model configuration as in TS-TCC: A token dimension of $64$ obtained from a linear projection with bias term, multi-head attention with four heads, and a total of four pre-norm transformer layers. The feedforward MLPs each consist of two layers with hidden dimensions $64$, with a ReLU activation and subsequent dropout layer in between and a final dropout layer at the end. Both dropout probabilities were set to 10%. We employed a transformer model (Vaswani et al., 2017) $\mathcal{S}$ to calculate the context vectors $c$, similar to TS-TCC. We observed in the supplementary implementation that Eldele et al. (2021) did not include positional encodings in their transformer tokens and thus used a set model instead of a

---

**Algorithm 1** The XIT method consists of the pretraining and finetuning phases.

---

**Input:** datasets for pretraining $D_{\text{PT}}$ (unlabeled) and finetuning $D_{\text{FT}}$ (labeled); batch size $B$; constants $\alpha$, $\beta$, and $\tau$; random strong and weak augmentations $A_s$ and $A_w$; initialized models $\mathcal{F}_{\boldsymbol{\theta}}$, $\mathcal{S}$, $g$, and $h$; gradient descent optimizer $O_{\text{PT}}$ over all four models; initialized classifier model $\mathcal{C}$; gradient descent optimizer $O_{\text{FT}}$ over $\mathcal{C}$ (and optionally $\mathcal{F}_{\boldsymbol{\theta}}$)
**Output:** learned models $\mathcal{F}_{\boldsymbol{\theta}}$ and $\mathcal{C}$

**if** no parameters $\boldsymbol{\theta}$ of $\mathcal{F}$ are known **then**
   # Pretraining phase
   **for** mini-batch $\{\boldsymbol{x}_i\}_{i=1}^B \sim D_{\text{PT}}$ **do**                       ▷ loop until convergence
      **for all** $i \in \{1, \ldots, B\}$ **do**
         # Projections
         $\boldsymbol{z}_i \leftarrow \mathcal{F}_{\boldsymbol{\theta}}(\boldsymbol{x}_i)$
         $\boldsymbol{\kappa}_i \leftarrow h\Big(\mathcal{S}\Big(\boldsymbol{z}_i^{1:(K-1)}\Big)\Big)$
         # XD-MixUp
         $\lambda_i \sim \text{Beta}(\alpha, \alpha)$
         $j \leftarrow (i+1) \mod B$
         $\widetilde{\boldsymbol{x}}_i \leftarrow \lambda_i \boldsymbol{x}_i + (1 - \lambda_i)\boldsymbol{x}_j$
         # Strong augmentation
         $\widetilde{\boldsymbol{x}}_{i,s} \sim A_s(\widetilde{\boldsymbol{x}}_i)$
         $\widetilde{\boldsymbol{z}}_{i,s} \leftarrow \mathcal{F}_{\boldsymbol{\theta}}(\widetilde{\boldsymbol{x}}_{i,s})$
         $\widetilde{\boldsymbol{c}}_{i,s} \leftarrow \mathcal{S}\Big(\widetilde{\boldsymbol{z}}_{i,s}^{1:(K-1)}\Big)$
         $\boldsymbol{\kappa}_{i,s} \leftarrow h(\widetilde{\boldsymbol{c}}_{i,s})$
         # Weak augmentation
         $\widetilde{\boldsymbol{x}}_{i,w} \sim A_w(\widetilde{\boldsymbol{x}}_i)$
         $\widetilde{\boldsymbol{z}}_{i,w} \leftarrow \mathcal{F}_{\boldsymbol{\theta}}(\widetilde{\boldsymbol{x}}_{i,w})$
         $\widetilde{\boldsymbol{c}}_{i,w} \leftarrow \mathcal{S}\Big(\widetilde{\boldsymbol{z}}_{i,w}^{1:(K-1)}\Big)$
         $\boldsymbol{\kappa}_{i,w} \leftarrow h(\widetilde{\boldsymbol{c}}_{i,w})$
      **end for**
      # Compute the Temporal Contrasting loss

$$\mathcal{L}_{\text{TC}}^s = -\frac{1}{B}\sum_{i=1}^B \log\left(\frac{g(\widetilde{\boldsymbol{c}}_{i,w}, \widetilde{\boldsymbol{z}}_{i,s}^K)}{\sum_{j=1}^B g(\widetilde{\boldsymbol{c}}_{i,w}, \widetilde{\boldsymbol{z}}_{j,s}^K)}\right)$$

$$\mathcal{L}_{\text{TC}}^w = -\frac{1}{B}\sum_{i=1}^B \log\left(\frac{g(\widetilde{\boldsymbol{c}}_{i,s}, \widetilde{\boldsymbol{z}}_{i,w}^K)}{\sum_{j=1}^B g(\widetilde{\boldsymbol{c}}_{i,s}, \widetilde{\boldsymbol{z}}_{j,w}^K)}\right)$$

      $\mathcal{L}_{\text{TC}} \leftarrow \frac{1}{2}(\mathcal{L}_{\text{TC}}^s + \mathcal{L}_{\text{TC}}^w)$
      # Compute the Soft Interpolation Contextual Contrasting loss
      Form $\mathfrak{B}^s = (\boldsymbol{\kappa}_{1,l}, \ldots, \boldsymbol{\kappa}_{B,l}, \boldsymbol{\kappa}_{1,s}, \ldots, \boldsymbol{\kappa}_{B,s}, \boldsymbol{\kappa}_{1,r}, \ldots, \boldsymbol{\kappa}_{B,r})$
      Form $\mathfrak{B}^w = (\boldsymbol{\kappa}_{1,l}, \ldots, \boldsymbol{\kappa}_{B,l}, \boldsymbol{\kappa}_{1,w}, \ldots, \boldsymbol{\kappa}_{B,w}, \boldsymbol{\kappa}_{1,r}, \ldots, \boldsymbol{\kappa}_{B,r})$
      Compute $\mathcal{L}_{\text{SICC}}(\mathfrak{B}^s)$ and $\mathcal{L}_{\text{SICC}}(\mathfrak{B}^w)$ according to eq. (3)
      $\mathcal{L}_{\text{SICC}} \leftarrow \frac{1}{2}(\mathcal{L}_{\text{SICC}}(\mathfrak{B}^s) + \mathcal{L}_{\text{SICC}}(\mathfrak{B}^w))$
      # Complete iteration
      $\mathcal{L}_{\text{Total}} \leftarrow \beta\mathcal{L}_{\text{TC}} + (1 - \beta)\mathcal{L}_{\text{SICC}}$
      Update all model parameters with $O_{\text{PT}}$ to minimize $\mathcal{L}_{\text{Total}}$
   **end for**
   Store learned parameters $\boldsymbol{\theta}$ of $\mathcal{F}$
**end if**

# Finetuning phase
**for** mini-batch $\{(\boldsymbol{x}_i, y_i)\}_{i=1}^B \sim D_{\text{FT}}$ **do**                   ▷ loop until convergence
   $\boldsymbol{z}_i \leftarrow \mathcal{F}_{\boldsymbol{\theta}}(\boldsymbol{x}_i)$
   $\widehat{\boldsymbol{y}}_i \leftarrow \mathcal{C}(\boldsymbol{z}_i)$                     ▷ Obtain class probabilities
   $\mathcal{L}_{\text{Class}} \leftarrow \text{CE}(\widehat{\boldsymbol{y}}_i, y_i)$              ▷ Use cross-entropy as criterion
   Update model parameters with $O_{\text{FT}}$ to minimize $\mathcal{L}_{\text{Class}}$
**end for**
**return** learned composed model $\mathcal{F}_{\boldsymbol{\theta}} \circ \mathcal{C}$

---

Table 4: This provides an overview of the datasets we used to evaluate and compare our method with the baselines. Note that we always only used the first variate in the case of multivariate datasets.

| Name | Train | # Samples Validation | Test | Length | # Classes | Balanced |
|---|---|---|---|---|---|---|
| Sleep-EDF | 25,612 | 7,786 | 8,910 | 3,000 | 5 | No |
| FD-A | 8,184 | 2,728 | 2,728 | 5,120 | 3 | No |
| HAR | 5,881 | 1,471 | 2,947 | 128 | 6 | No |
| ECG | 43,673 | 10,920 | 1,904 | 1,500 | 4 | No |
| Epilepsy | 7,360 | 1,840 | 2,300 | 178 | 2 | No |

Table 5: The distribution of datasets from the UCR repository used in our experiments.

| Domain | Sequence Length min | mean | max | Dataset count | Train Size min | median | max |
|---|---|---|---|---|---|---|---|
| Audio | 270 | 337.5 | 405 | 2 | 60 | 132 | 204 |
| Device | 96 | 120.0 | 144 | 2 | 180 | 4553 | 8926 |
| Ecg | 82 | 113.5 | 140 | 4 | 23 | 61 | 500 |
| Eeg | 50 | 230.0 | 510 | 5 | 56 | 316 | 5890 |
| Har | 30 | 127.2 | 315 | 9 | 30 | 151 | 2238 |
| Image | 23 | 226.1 | 512 | 30 | 16 | 399 | 81714 |
| Meg | 200 | 200.0 | 200 | 1 | 727 | 727 | 727 |
| Motion | 8 | 229.8 | 343 | 14 | 36 | 332 | 7494 |
| Other | 36 | 118.5 | 201 | 2 | 18 | 1238 | 2459 |
| Sensor | 24 | 273.1 | 577 | 14 | 20 | 85 | 3636 |
| Simulated | 15 | 159.4 | 500 | 8 | 20 | 93 | 1000 |
| Sound | 217 | 217.0 | 217 | 1 | 3315 | 3315 | 3315 |
| Spectro | 234 | 382.0 | 570 | 7 | 28 | 57 | 613 |
| Traffic | 24 | 24.0 | 24 | 1 | 20 | 20 | 20 |
| Total | 8 | 221.1 | 577 | 100 | 16 | 180 | 81714 |

sequence model. Nevertheless, we decided not to use it either since adding a sine-cosine positional encoding (Vaswani et al., 2017) did not noticeably affect the results.

**Finetuning**   To evaluate the utility of the learned representation for classification, we trained simple classifiers $\mathcal{C}$ on top of all encoder models $\mathcal{F}_{\theta}$. This is embedded into the complete procedure as shown in Algorithm 1. Following the *linear probing* experiment of van den Oord et al. (2018) for good comparability, a neural classifier head with a single linear layer is trained while the encoder is frozen. The final output is transformed with a softmax, and the training criterion is cross-entropy. The hyperparameters for finetuning are given in Table 6.

**Details on TF-C**   We changed the frequency transformation via the FFT to be orthonormal by scaling the result of length $T$ by $\sqrt{T}$. This preserves the signal's magnitude, allowing us to perform training and inference in 16-bit mixed precision without numerical issues. Furthermore, we use the complete pretraining datasets instead of only subsets for the N-to-one settings (Zhang et al., 2022, Appendix K). We use mostly the same hyperparameters as in Appendix E when pretraining TF-C. However, we deviate slightly to follow the linear probing evaluation. This means that in finetuning, we only train a single-layer classifier head instead of a deeper MLP. We only optimize the classifier and classifier loss instead of training the encoder as well or optimizing the pretraining and classifier losses jointly. We use the very same encoder config for all datasets for a more direct comparison, especially in the multi-dataset experiments.

### A.3   Additional Results

First, we show how our approach compares to the baselines when run on increasing fractions of the labeled target datasets. We follow the same setup as in Table 1, so all experiments were run five times

Table 6: This table shows the differing hyperparameters we chose by search using *Optuna* (Akiba et al., 2019). In particular, we also considered larger batch sizes of up to 1,024 for the pretraining phase but did not find them to be beneficial, much like in the works of TS-TCC and TF-C.

| Model | Pretraining | | | Finetuning | | |
|---|---|---|---|---|---|---|
| | Batch size | LR | Weight decay | Batch size | LR | Weight decay |
| XIT | 64 | 0.0001 | 0.0003 | 64 | 0.00014 | 0.0016 |
| TS-TCC | 128 | 0.0003 | 0.0003 | 64 | 0.00014 | 0.0016 |
| TF-C | 64 | 0.0003 | 0.0005 | 64 | 0.0003 | 0.0003 |
| TS2Vec | 16 | 0.001 | 0.0005 | 64 | 0.00014 | 0.0016 |
| TNC | 64 | 0.001 | 0.0005 | 64 | 0.00014 | 0.0016 |
| T-Loss | 20 | 0.001 | 0.0005 | 64 | 0.00014 | 0.0016 |
| Supervised | – | – | – | 64 | 0.00014 | 0.0016 |

with different seeds. Results are shown in Figure 5. We excluded Epilepsy since, except for TF-C, all models performed similarly well for 5% of the data and more. We can generally conclude that our method works best in low-data scenarios and has the smallest variance. We also suspect that some bad data samples are present in ECG since increasing the amount of data caused a deterioration in classification performance. Similar effects might affect TF-C when pretrained on Sleep-EDF.

Table 1 in the main paper only contained AUROC scores due to space constraints. Thus, we provide Accuracy and Macro F1 scores for easier comparison in Table 8 and Table 7, respectively.

As a supplement to the inspection of the embedding spaces in Section 3.3, we provide a qualitative visual excerpt of their low-dimensional projections on three datasets. See Figure 6.

Table 7: Macro F1 scores in percent for the evaluation shown in Table 1. Higher is better.

| PT | Model | Sleep-EDF | FD-A | HAR | ECG | Epilepsy |
|---|---|---|---|---|---|---|
| 0 | Superv. | 28.36 ±4.8 | 70.36 ±3.8 | 36.20 ±2.3 | **24.09 ±1.4** | 79.17 ±1.5 |
| 1 | XIT | 29.88 ±1.1 | 80.56 ±0.8 | **37.85 ±3.3** | 22.65 ±0.9 | 71.61 ±15.9 |
| | TS-TCC | **35.30 ±0.9** | 20.84 ±0.0 | 5.10 ±0.0 | 21.97 ±0.8 | 45.22 ±1.8 |
| | TF-C | 34.77 ±1.6 | 67.18 ±9.7 | 24.85 ±5.1 | 17.68 ±0.8 | 44.46 ±0.0 |
| 2 | XIT | 31.30 ±1.2 | 81.98 ±1.3 | 37.02 ±2.0 | 22.81 ±0.5 | **83.84 ±1.9** |
| | TS-TCC | 33.57 ±1.6 | 20.84 ±0.0 | 6.34 ±2.6 | 22.98 ±0.6 | 44.43 ±0.0 |
| | TF-C | 29.62 ±5.9 | 56.83 ±7.8 | 29.18 ±3.6 | 17.39 ±0.4 | 44.46 ±0.0 |
| 3 | XIT | 32.57 ±1.7 | **82.35 ±1.4** | 35.48 ±3.0 | 23.49 ±0.7 | 82.22 ±1.9 |
| | TS-TCC | 30.77 ±2.2 | 20.84 ±0.0 | 6.74 ±1.7 | 22.89 ±0.8 | 44.43 ±0.0 |
| | TF-C | 29.64 ±3.9 | 74.66 ±5.0 | 28.94 ±9.0 | 22.23 ±1.4 | 44.46 ±0.0 |
| 4 | XIT | 28.33 ±2.1 | 77.57 ±2.7 | 33.88 ±1.7 | 24.01 ±1.7 | 80.18 ±7.9 |
| | TS-TCC | 26.71 ±1.7 | 20.84 ±0.0 | 13.32 ±2.0 | 23.00 ±0.5 | 44.43 ±0.0 |
| | TF-C | 29.95 ±3.6 | 71.89 ±2.4 | 27.83 ±7.6 | 19.45 ±3.0 | 44.46 ±0.0 |

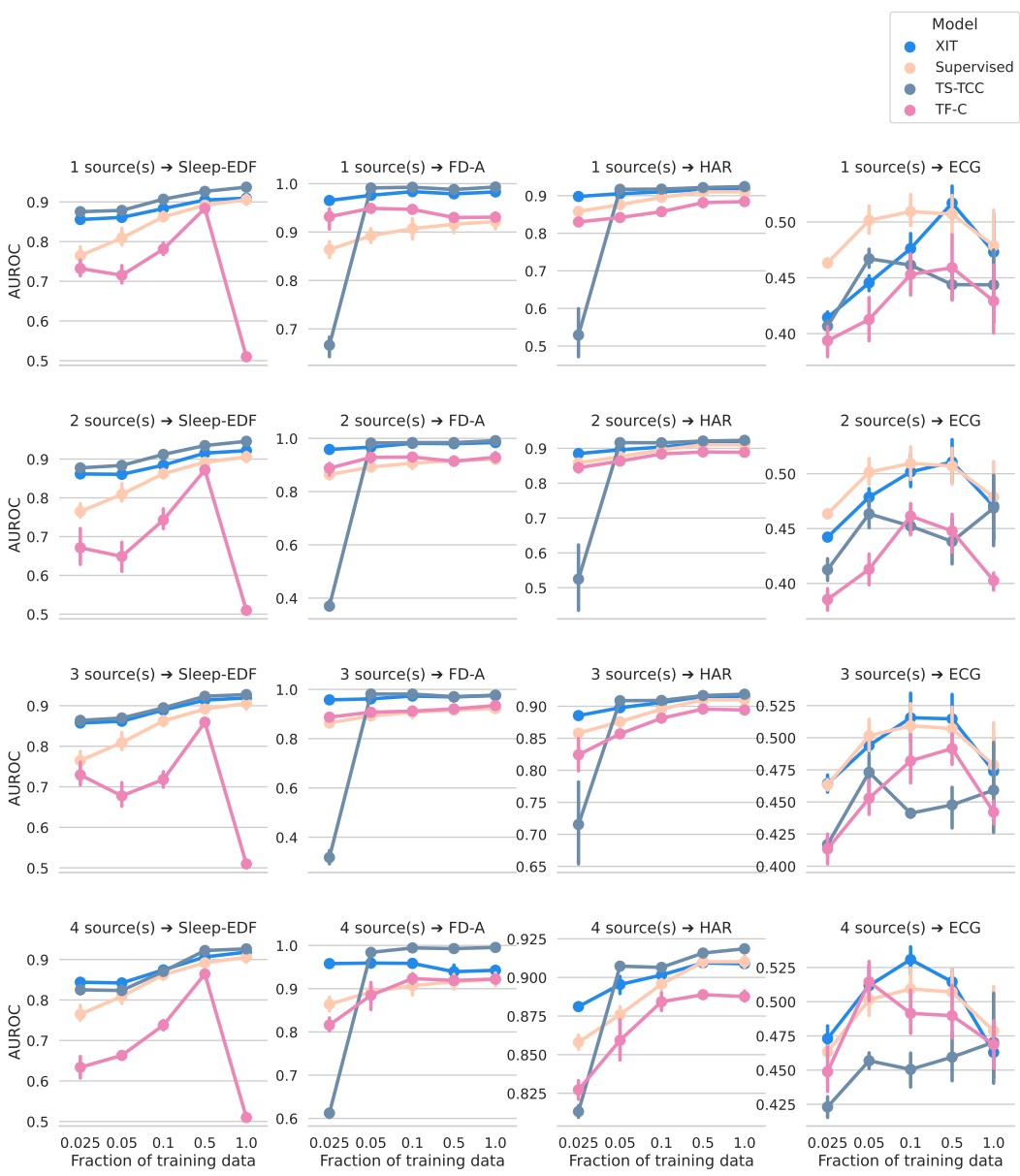

Figure 5: Comparison of XIT to TS-TCC, TF-C, and supervised when pretraining on one to four datasets and subsequent finetuning to each. The performance is measured in AUROC, where higher is better. Please note the differently scaled y-axes.

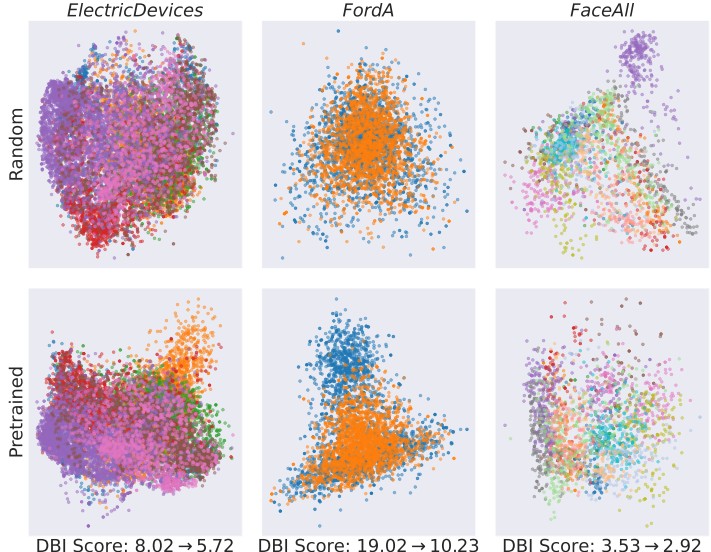

Figure 6: Qualitative excerpt from a single pretrained XIT model trained on 75 UCR datasets, evaluated on three hold-out datasets. The figure was generated by performing a reduction to two dimensions via Principal Component Analysis on the output of the encoder $\mathcal{F}$. Furthermore, we include the DBI to further support our visual observation. Lower DBI $\rightarrow$ more separated clusters.

Table 8: Accuracy scores in percent for the evaluation shown in Table 1. Higher is better.

| PT | Model | Sleep-EDF | FD-A | HAR | ECG | Epilepsy |
|---|---|---|---|---|---|---|
| 0 | Superv. | 45.92 ±3.9 | 71.28 ±3.1 | 44.38 ±2.2 | 47.98 ±1.5 | 89.52 ±0.6 |
| 1 | XIT | 55.33 ±1.0 | 77.72 ±1.0 | **46.07 ±2.3** | 45.10 ±1.8 | 87.48 ±4.9 |
| | TS-TCC | 57.06 ±0.7 | 45.46 ±0.0 | 18.07 ±0.0 | 50.10 ±0.6 | 80.12 ±0.3 |
| | TF-C | **57.53 ±1.0** | 65.91 ±9.6 | 38.46 ±3.3 | 51.81 ±0.2 | 80.04 ±0.0 |
| 2 | XIT | 54.70 ±1.2 | 79.21 ±1.7 | 45.48 ±1.1 | 46.88 ±0.6 | **91.38 ±0.8** |
| | TS-TCC | 56.17 ±1.1 | 45.46 ±0.0 | 18.25 ±0.3 | 49.20 ±1.4 | 79.96 ±0.0 |
| | TF-C | 50.71 ±5.8 | 56.51 ±5.2 | 39.23 ±2.2 | **51.82 ±0.1** | 80.04 ±0.0 |
| 3 | XIT | 54.69 ±1.0 | **79.56 ±1.7** | 43.77 ±1.7 | 47.40 ±1.6 | 90.69 ±0.8 |
| | TS-TCC | 54.76 ±1.5 | 45.46 ±0.0 | 18.06 ±0.8 | 50.31 ±0.3 | 79.96 ±0.0 |
| | TF-C | 53.47 ±4.8 | 70.29 ±5.9 | 38.65 ±8.3 | 48.14 ±4.1 | 80.04 ±0.0 |
| 4 | XIT | 50.91 ±1.5 | 74.20 ±2.8 | 43.18 ±1.1 | 46.89 ±2.2 | 90.11 ±3.0 |
| | TS-TCC | 50.74 ±1.8 | 45.46 ±0.0 | 24.67 ±1.5 | 49.33 ±0.8 | 79.96 ±0.0 |
| | TF-C | 49.18 ±3.9 | 67.38 ±2.7 | 39.24 ±5.8 | 51.37 ±0.4 | 80.04 ±0.0 |

