# OpenReview forum: "United We Train, Divided We Fail! Representation Learning for Time Series by Pretraining from 75 Datasets at Once"
_ICLR.cc/2024/Conference — Submitted to ICLR 2024_

### Official Review · Reviewer_4uSe · 2023-11-01

**Soundness:** 4 excellent
**Presentation:** 4 excellent
**Contribution:** 3 good
**Rating:** 6
**Confidence:** 4

**Summary:**

Authors propose a self-supervised pre-training method for time-series data encompassing various domains and temporal dynamics and show that it leads to significant performance gains for finetuning on small datasets. The problem is well-motivated as time-series occurs in domains such as traffic, weather, finance, etc and in many cases labelled data is limited.

Given a time-series $x \in R^T$ authors propose to encode it as a sequence $z \in R^{K \times d}$ via a simple convnet. For this, authors collect various labelled and unlabelled time-series datasets (>75) and pretrain on all of them. Pretraining consists of various objectives from prior works and some novel components. At a high level, the encoder is trained to predict $z_K$ from the context $z[<K]$. During finetuning $z_K$ is used to classify $x$. For better generalization, during pretrianing, a different time-series $y$ is chosen from the batch and a convex combination $\lambda x + (1-\lambda)y$ is used for learning a continuous latent space between samples from various datasets. Training losses are designed to learn representations that are robust to augmentations such as magnitude scaling, permutation and jitter.

Authors demonstrate the utility of the method on a large collection of timeseries datasets over various modalities.

**Strengths:**

1. The experimental evaluation is comprehensive and the empirical gains over supervised training are significant, clearly demonstrating the utility of the proposed method for prertaining on diverse datasets.

**Weaknesses:**

1. The novelity of the method is limited and the main components such as augmentations and TC loss are borrowed from previous works such as TS-TCC.

2. The significance of the novel contribution, SICC loss, isnt made fully clear. In Table 1 the experiments are performed on a limited number of datasets and data, and it seems that including the SICC loss helps. In contrast, Table 2 and Figure 6 suggets that finetuning on all available data, excluding the SICC loss (essentially resulting in the TS-TCC model from prior work) doesnt hurt the performance significantly.

3. The TS-TCC framework is itself not intuitive to begin with - what exactly is the need for such data augmentations and contrived losses derived from these? Why to only use the last vector $z_K$ to be predicted from context $c$ - why not predict $z_t$ from $z_{t-1}$ for every $t$? How is $c$ formed? Not clear if this pipeline cannot be replaced with a much simpler training pipeline assuming more data is available.

**Questions:**

Typo: pg 6 "in contrary" -> "in contrast"

---

> ### Author Response · Authors · 2023-11-16
>
> We thank you for your very thoughtful review and for pointing out positive aspects as well as open opportunities, which we try to address in the following.
>
> We agree that we provided a very extensive overview of current approaches and their comparability to supervised learning.
>
> Your raised points have greatly helped us to improve our manuscript.
> We address all your concerns below.
>
> > XIT only provides minor novelty over TS-TCC.
>
> XIT's core encoder and TC module are derived from TS-TCC. We substitute its previous NT-Xent loss, integrating multiple runs of their TC loss. This configuration solely employs the cross-prediction task, presenting entirely novel inputs and outputs. Dismissing its novelty would be akin to disallowing the use of a Convolutional ResNet as a backbone in a new model. The underlying concept emphasizes the explicit need for both intra- and inter-dataset structuring tasks simultaneously.
>
> The application of mixup on time series by Wickstrøm et al. (2022), on the other hand, is mainly a data augmentation method with the goal of learning a representation by predicting the mixing lambda. This is challenging for intra-dataset pretext tasks. For inter-dataset tasks, the model tends to learn to differentiate between datasets, which is not the goal of this task. We tackle this by ensuring that the lambda is not simply predicted but that the latent space corresponds to the time domain in terms of mixing distance. This ensures that we can really leverage cross-dataset information. Our experimental evaluation shows that this intuition carries over into practice.
> As a side note: In initial probings, we did not see any meaningful effect by simply predicting the lambda as proposed by Wickstrøm et al. (2022).
>
> > The significance of the SICC loss is not made clear in the ablations
>
> Thank you for pointing that out. We updated the paper to make this more clear, see Table 3 (Pg. 8). The evaluation indeed shows that we need all of our proposed methods to improve the performance. More details are in the Global Response 3.
>
> > The TS-TCC framework is itself not intuitive to begin with - what exactly is the need for such data augmentations and contrived losses derived from these?
>
> The need for weak and strong augmentations is already extensively discussed in the TS-TCC paper (see section 3.1 and ablations in section 5.4). The general motivation is that for a model to learn strong representations, one needs to pose a real challenge by means of the loss formulation. This is exactly what TS-TCC does, as predicting the weak from the strong augmented variant (and vice versa) can indeed be challenging. We, however, agree that simpler models should also be studied in further work.
>
> >  Is it possible to predict z_t from z_t-1 for every t?
>
> While it is possible, it would also increase the required memory tremendously. The SICC loss uses the predicted context of all elements in the batch. Building the required distance matrix of all pairs is fairly expensive, which one likely would not want to repeat for each time window. Furthermore, it is sometimes likely not an expressive task for time series with lots of leading padding.
>
> > Is it possible to replace this pipeline with a much more simple one?
>
> Please see the second part of our answer to question one. However, in essence, simple mixup was not enough, and this is why we wanted to induce specific latent structuring methods for this specific challenge. However, making it easier is an interesting course of action and will be on our list of future work.
>
> >  How is the context vector formed?
>
> We utilize a sequence transformer (summarization model $\mathcal{S}$) taking the previous $z_k$, where $k \in [0, t-1]$ as input and producing a “forecast” of the context vector (see section 2.2). To get a more visual explanation, please see Figure 2 of our paper. For implementation details, please refer to “Encoder & Summarization Model” in appendix A.2.
>
> We hope that we have clarified your concerns and that you can reconsider our rating. Do let us know if you have any further questions or comments.

---

> > ### Comment · Reviewer_4uSe · 2023-11-17
> >
> > I thank the authors for the detailed and convincing clarifications - the updated tables look clearer. I'm maintianing my score as I find the contrbutions made by the authors to be non-trivial. While I'm not fully certain if the method has a wide-enough appeal (and I might be completely wrong), I'll be supporting it during the discussion phase among the reviewers.
> > In case the paper isn't accepted, I urge the authors to work towards simplifying the writing (and the method itself if possible) and resubmit to a different venue as the reviews might have been overtly harsh - I believe the paper deserves better scores.

---

> > > ### Author Response · Authors · 2023-11-19
> > >
> > > Thank you for your thoughtful answer and continued support. We're pleased that the updated tables clarified the presentation.
> > >
> > > Your recognition of the non-trivial contributions is appreciated. In response to your point about broader appeal, we note the growing importance of Self-supervised Representation Learning on Time series. Our method aligns with this trend, aiming to unlock the potential of more data, inspired by successful strategies in vision and NLP.
> > >
> > > We also want to highlight recent work in forecasting, such as TimeGPT (https://arxiv.org/pdf/2310.03589.pdf). This study illustrates the significance of time series pretraining, emphasizing its necessity for multiple scenarios. While TimeGPT leans towards a more heavyweight (large transformer) computational approach and is, as far as we know, closed source (nixtla plans monetarization), our framework is designed to be accessible and adaptable without the same computational demands.
> > >
> > > We appreciate your feedback on potential simplifications and, if needed, are committed to refining the writing and method for resubmission to a different venue. We share your optimism about the paper's potential.
> > >
> > > Thanks again for your time and insights.

---

### Official Review · Reviewer_Gxh9 · 2023-11-01

**Soundness:** 2 fair
**Presentation:** 3 good
**Contribution:** 2 fair
**Rating:** 3
**Confidence:** 4

**Summary:**

The paper summarizes a way to pre-train a model on multiple datasets to learn representations which are useful for downstream tasks such as classification. The authors propose modifications to an earlier method called TS-TCC, which introduces weak and hard augmentations to time-series. The authors propose a novel loss function SICC which ensures that augmented time-series contexts are similar to the time-series contexts from which they are interpolated.

**Strengths:**

1. The paper is well written and easy to follow.
2. To the best of my knowledge, pre-training on multiple time-series datasets has not been explored.
3. I like the vision behind the experimental section, as well as its organization into research questions.

**Weaknesses:**

1. **Contributions:** It is unclear how significant the contributions of the study are beyond training on multiple datasets. It seems to be an incremental improvement over TS-TCC and the experimental setup of TF-C. Could the authors experimentally demonstrate why it is infeasible to use TS-TCC (and TF-C) for pre-training on multiple datasets?
2. **Experimentation:** I like how the authors have structure their experimentation in terms of research questions. However, I feel that the results are not convincing due to several reasons: (1) Multiple baselines including recent works such as TS2Vec [1], older techniques such as T-Loss [2] and TST [3], and statistical methods such as Dynamic Time Warping-based Nearest Neighbors (see [1]) were missing. (2) The experiment framework differs from some prior work, where the representations are used to train a downstream classifier such as SVM for classification (see [1] and [2] for example). I wonder whether, how and why is the fine-tuning and subsequent evaluation different from that of prior work, including the metrics used to compare the methods (i.e. accuracy). Also see section on Clarity below. (3) The ablation results in Table 2 seem insignificant since the variances of the models are overlapping. I would be interested in seeing a critical difference diagram to see if the differences between the proposed method and ablations are indeed significant.
3. **Clarity:** Some important details in experimentation were missing, for example, how were the models fine-tuned? I see from the appendix that cross-entropy loss was used, but it is unclear what was the structure of the model.

References:
[1] Yue, Zhihan, et al. "Ts2vec: Towards universal representation of time series." Proceedings of the AAAI Conference on Artificial Intelligence. Vol. 36. No. 8. 2022.
[2] Franceschi, Jean-Yves, Aymeric Dieuleveut, and Martin Jaggi. "Unsupervised scalable representation learning for multivariate time series." Advances in neural information processing systems 32 (2019).
[3] Zerveas, George, et al. "A transformer-based framework for multivariate time series representation learning." Proceedings of the 27th ACM SIGKDD conference on knowledge discovery & data mining. 2021.

**Questions:**

1. How are the models fine-tuned?
2. Could you please report critical difference diagrams of fine-tuning performance on all datasets? Could you show the impact of pre-training, as how sample efficient the fine-tuning procedure is?
3. Also see questions above.

---

> ### Author Response · Authors · 2023-11-16
> **Comment by the Authors**
>
> The authors thank the reviewer for the very extensive review, highlighting strengths and raising opportunities for improving our manuscript. This really helped us to improve upon our previous manuscript and helped to provide an even clearer scientific story.
>
> We address all your concerns below.
>
> > Could the authors experimentally demonstrate why it is infeasible to use TS-TCC (and TF-C) for pre-training on multiple datasets?
>
> We provided an intuition in Table 1, but in addition to that, we made a much more extensive experimental evaluation in Table 2. Next to even other baselines, it illustrates the superiority of XIT compared to the other methods.
>
> > Multiple standard baselines are missing.
>
> We thank you for pointing that out. We provided 3 more baselines beyond TS-TCC and TF-C, as shown in the newly added Table 2. Namely, we evaluate TS2Vec (Yue et al., 2022), TNC (Tonekaboni et al., 2020), and T-Loss (Franceschi et al., 2019):
>
> | Target Dataset     		 | XIT      		 | TS-TCC   		 | TF-C      		 | TS2Vec   		 | TNC      		 | T-Loss   		 |
> |-----------------------------|-------------------|-------------------|--------------------|-------------------|-------------------|-------------------|
> | Average Macro F1 $\uparrow$ | **54.4±3.2** | 51.1±3.5 		 | 43.1±6.1  		 | 53.7±2.5 		 | 36.4±9.1 		 | 34.2±4.7 		 |
> | Rank $\downarrow$  		 | **2.10±1.2** 		 | 3.06±1.6 		 | 3.56±1.8  		 | 2.82±1.6 		 | 4.48±1.0 		 | 4.98±1.1 |
>
> The results strengthen the empirical effectiveness of our method.
>
> > The experiment framework differs from some prior work, where the representations are used to train a downstream classifier such as SVM for classification.
>
> While TS2vec indeed follows a slightly different scheme, we adhere to the standard linear benchmarking evaluation approach by Oord et al. (2018) and Chen et al. (2020), consistent with the evaluation methodology of TS-TCC and TF-C. In particular, we train a linear classifier (single MLP layer) on top of a frozen self-supervised pretrained encoder model.
>
> > Why didn’t you include accuracy in your evaluation?
>
> We provided exemplary accuracy scores in the appendix for Table 1. However, especially with unbalanced datasets (UEA + low data sampling), accuracy may not be a reliable metric. Therefore, in line with TS-TCC, we also presented AUROC and Macro-F1 scores.
>
> > What was the structure of the model?
>
> Refer to our response in question 3. Additionally, we have adjusted the manuscript for improved clarity.
>
> > The ablation results seem insignificant.
>
> Thank you for pointing that out. The ablation results were indeed not presented in full clarity. To improve, we revamped our results in the new Table 3 by applying ranking after Demšar et al. (2006).
> | Pretraining Component  	| AUROC rank $\downarrow$ | Accuracy rank $\downarrow$ | Macro F1 rank $\downarrow$ |
> |----------------------------|-------------------------|----------------------------|----------------------------|
> | XD-MixUp + SICC + TC (XIT) | **1.800 ±0.91**	| **1.780 ±0.89**   	| **1.780 ±0.84**   	|
> | XD-MixUp + SICC        	| 3.560 ±0.87         	| 3.700 ±0.56            	| 3.580 ±0.79            	|
> | XD-MixUp + TC          	| 2.060 ±0.94         	| 1.920 ±0.84            	| 1.980 ±0.91            	|
> | TC                     	| 2.580 ±0.91         	| 2.600 ±0.78            	| 2.660 ±0.81            	|
>
> > How are the models fine-tuned?
>
> For the experiments in Table 1, we fine-tuned on 2.5% of the data. For the subsequent experiments, we fine-tuned on 100% of the hold-out dataset since many UCR datasets are already challengingly small.
>
> We trained for 2000 steps with early stopping. For details, we provide a paragraph on “Implementation details” in Appendix A.2.
>
> For the best possible outcome, we run a thorough hyperparameter search for the optimal classifier config. Finally, we ran the experiments on multiple seeds.
>
> >  How sample efficient is the fine-tuning procedure?
>
> This is the main challenge that we want to tackle with this paper. We particularly leverage the structures of multiple datasets in situations when large amounts of labeled data are not available. Therefore, the finetuning procedure is very sample efficient. Namely, the results in Table 1. were obtained from fine-share
> tuning on as little as 147 time series (HAR).
>
> We hope that we have clarified your concerns and that you can reconsider our rating. Do let us know if you have any further questions or comments.

---

> ### Author Response · Authors · 2023-11-20
> **Looking Forward to Feedback on Our Response**
>
> Dear Reviewer,
>
> We appreciate the time and effort that you have taken to provide us with your insightful review. We would like to ask if the reviewer has any further concerns or is satisfied by our responses to the original review.
>
> We are looking forward to any further discussion with the reviewer and would like to thank the reviewer again for helping make our paper better.
>
> Regards,
> The Authors

---

> > ### Comment · Reviewer_Gxh9 · 2023-11-22
> > **Thanks for the rebuttal!**
> >
> > Dear Authors,
> > I appreciate the rebuttal. Thanks for answering some of my questions. I do believe that the paper is stronger given the comparison with TS2Vec and other methods.
> >
> > Quick question: What does "we fine-tuned on 100% of the hold-out dataset" mean? I am assuming you are referring to 100% of the training set.
> >
> > Also I apologize for the late response, but it is hard to follow changes in the paper if they are not done in a different ink.
> >
> > My request on using critical difference diagrams for comparisons was not answered -- I would like to note that comparing models via critical difference diagram is common practice in this field. There are some other questions and recommendations which were also not answered, for e.g., comparison with dynamic time warping and nearest neighbors methods.
> >
> > Given these, and the reviews from the other reviewers, I am inclined to stick with my current score. If the experimentation was made stronger, I would have been willing to increase my score.

---

> > > ### Author Response · Authors · 2023-11-22
> > >
> > > Dear Reviewer,
> > >
> > > Thank you for your response. Regarding your first question, during the pretraining procedure, the model did not encounter any labels. In the "evaluation" step, we finetuned on datasets that the model had not previously seen. For example, we pretrained on ArrowHead and Medical Images (only time series, no labels) and then finetuned on GunPoint (time series + labels). This is in contrast to the TFC-Experiments, where we finetuned on 2.5% of the target domain. For instance, we pretrained on HAR and Sleep-EDF (only time series, no labels) and finetuned on 2.5% of Epilepsy (time series + labels).
> > >
> > > Indeed, incorporating color to highlight the changes would have been more beneficial. Thank you for the hint.
> > >
> > > Regarding the CD diagrams, you're correct; we could have included those, but we opted for Ranks, which should convey the same information. We introduced three additional Baselines and decided against DTW and KNN, as they had already been evaluated against Ts2Vec. Additionally, it is not feasible to pretrain them, as KNN, by nature, is not trained. Furthermore, especially KNN, is not very computationally efficient with respect to the number of added data points.
> > >
> > > From our perspective, we have addressed all other open issues raised by the reviewers.

---

### Official Review · Reviewer_kMWa · 2023-11-01

**Soundness:** 3 good
**Presentation:** 3 good
**Contribution:** 1 poor
**Rating:** 3
**Confidence:** 4

**Summary:**

The paper challenges the common belief that pretraining is ineffective for time series data due to source-target mismatch. The authors introduce a novel approach called XIT, combining XD-MixUp, SICC, and Temporal Contrasting, to create a shared latent representation from up to 75 diverse unlabeled time series datasets, which outperforms supervised training and other self-supervised methods, especially in low-data scenarios. The work demonstrates the feasibility and effectiveness of multi-dataset pretraining for time series, debunking the prevailing myth and paving the way for further advancements in leveraging multiple datasets for improved time series classification and analysis.

**Strengths:**

The paper excels in its clear and effective communication of key concepts, ensuring a coherent and easily-followed narrative. It tackles a significant and intriguing challenge within time series analysis: the development of a pre-trained model by leveraging multiple diverse datasets. Furthermore, the paper rigorously examines the individual components of the proposed method through ablations, providing valuable insights into how each element influences downstream performance.

**Weaknesses:**

1. The novelty of the proposed method is constrained, as it heavily relies on TS-TCC (Eldele et al., 2021). The SICC loss, derived from previous works (Sohn, 2016) and (Chen et al., 2020), bears resemblance to the one used in TS-TCC and essentially facilitates soft alignments. While the authors claim XD-MixUp as a novel contribution, it closely resembles the mixup data augmentation scheme introduced by (Zhang et al., 2018) and previously applied in the time series domain by (Wickstrøm et al., 2022).
2. The method's evaluation is limited in scope, as it only compares against two pre-trained methods, one of which is closely related to the proposed approach. Notably, the paper overlooks significant baseline models such as TS2Vec (Yue et al., 2022), CoST (Woo et al., 2022a), and One Fits All (OFA) (Zhou et al., 2023).

**Questions:**

Is there any particular reason why some recent methods, like the ones described above, were not presented as baselines in the paper?

---

> ### Author Response · Authors · 2023-11-16
>
> Thank you for reading our paper and pointing out its strengths and opportunities for improvement.
>
> We are pleased to see you agree that the topic is highly relevant and exciting, that the paper is written clearly, and that you find the ablation studies sound. We further improved the presentation of the ablation study to make the results clearer to interpret and even more convincing (see Global Response 3). We answer your concerns next.
>
> > More recent baselines
>
> Thank you for bringing this up. We have now added many more recent baselines in the revised manuscript for a fair comparison.
>
> In particular, we follow your suggestion to compare with TS2Vec (Yue et al., 2022). However, we use different models than CoST (Woo et al., 2022a) and OFA (Zhou et al., 2023). The first is exclusively introduced as a forecasting model, whereas we focus on classification. The latter was released after the paper submission, and we therefore exclude it and will happily investigate it in future work. Instead, we first evaluated the already included TF-C (Zhang et al., 2022) and TS-TCC (Eldele et al., 2021) models on the UCR dataset, each across three seeds and with their respective ranks.
>
> Going even further, *we added three other self-supervised time series pretraining models* that are typically used for classification. Namely, we compared to TS2Vec (Yue et al., 2022), TNC (Tonekaboni et al., 2020), and T-Loss (Franceschi et al., 2019). Please see the Global Response (2. part) for the table of results of XIT and the in total five baselines.
>
> We can see that in light of this significantly extended comparison, our method performs favorably and further supports the necessity of our method.
>
> >  Novelty
>
> We discussed this in general in the Global Response (1.). Specifically regarding your assessment, we want to point out that Sohn (2016) and Chen et al. (2020) laid the groundwork for a large family of different pertaining methods, of which XIT is indeed a member. This, however, makes it no less novel, especially given that soft alignments, as in SICC, are not typically used there.
>
> Regarding the novelty of XD-MixUp, we want to point out that we reference both the original mixup idea by (Zhang et al., 2022) and the work of  Wickstrøm et al. (2022) in section 2.1. To contextualize it further: XD-MixUp serves a very different purpose than the variant of MixUp in both publications. In the original one, it was introduced as a data augmentation and for soft labels in classification tasks. We, however, employ it to structure the (intermediate) latent space and effectively connect separated datasets into a coherent shared representation. Wickstrøm et al. (2022) proposed a completely separate learning approach with a different loss that promotes correctly predicting the mixing factor $\lambda$. However, this does not explicitly structure the latent space and instead relies on its implicit effects. In the initial probings of our research, we did not find this approach to be working in the face of the more challenging multi-dataset setting we tackle.
>
> We hope that we have clarified your concerns and that you can reconsider our rating. Do let us know if you have any further questions or comments.

---

> ### Author Response · Authors · 2023-11-20
> **Looking Forward to Feedback on Our Response**
>
> Dear Reviewer,
>
> We appreciate the time and effort that you have taken to provide us with your insightful review. We would like to ask if the reviewer has any further concerns or is satisfied by our responses to the original review.
>
> We are looking forward to any further discussion with the reviewer and would like to thank the reviewer again for helping make our paper better.
>
> Regards,
> The Authors

---

### Official Review · Reviewer_KhAg · 2023-11-06

**Soundness:** 2 fair
**Presentation:** 3 good
**Contribution:** 2 fair
**Rating:** 3
**Confidence:** 3

**Summary:**

The paper proposes a new pretraining procedure for time-series datasets. It leverages two ideas in prior work, namely point wise MixUp and Temporal Contrastive loss. On top of these ideas the authors propose to adapt the sample contrastive loss from TS-TCC to take into account the sampled MixUp ratio. Experiments on common time series datasets show that the method outperforms the very related TS-TCC baseline. Moreover, additional experiments with multiple datasets show that there is an improvement the more datasets that are used and ablation studies on each component of the proposed method show that they contribute towards a better performance.

**Strengths:**

The idea is well presented and makes use of two methods that have shown to provide significant improvements in time-series modeling.

The paper has a thorough experimental setup with multiple datasets that are popular in this field. In addition, the authors perform multiple runs and report the standard deviation for all experiments.

**Weaknesses:**

Given that this paper proposes a relatively small change in prior work the experimental results need to be strong and clearly show that this change provides an improvement over the baseline. However, for the first experiment which is the same setting as TS-TCC, the paper firstly reports a different metric and secondly significantly lower performance than the one reported in the TS-TCC paper. Moreover, table 1 reports AUROC significantly <0.5 for Epilepsy which is a binary classification problem.

One of the main premises of the paper is that this method allows pretraining on multiple datasets. It is that the experiment on the UCR datasets does provide some improvement when using more datasets for pretraining. However, the standard deviations are so high that it makes the result very minor. Moreover, there is no evidence provided that this would not apply to TS-TCC. On the contrary, in table 1 TS-TCC seems to improve with more datasets more often than XIT.

Finally the standard deviation in the ablation study is so large that it makes it hard to draw conclusions regarding the benefit of the components of the method. For instance one could argue that MixUp and TC are as good as XIT.

**Questions:**

What is the performance of XIT for table 1 using the same setting as TS-TCC for their table 2 and fig 3?

How can the area under the ROC curve for binary classification be ~0.2 on table 1 last column for TF-C?

---

> ### Author Response · Authors · 2023-11-16
> **Comment by Authors**
>
> Thank you for the extensive review and depicting the strong points as well as pointing out opportunities for improvement.
>
> Your raised points have greatly helped us to improve our manuscript. We address your concerns below.
>
> >  XIT only provides minor novelty over TS-TCC and MixUp.
>
> XIT's core encoder and TC module are derived from TS-TCC. We substitute its previous NT-Xent loss, integrating multiple runs of their TC loss. This configuration solely employs the cross-prediction task, presenting entirely novel inputs and outputs. Dismissing its novelty would be akin to disallowing the use of a Convolutional ResNet as a backbone in a new model. The underlying concept emphasizes the explicit need for both intra- and inter-dataset structuring tasks simultaneously.
>
> Application of mixup on time series by Wickstrøm et al. (2022) is mainly a data augmentation method with the goal of learning a representation by predicting the mixing $\lambda$. This is challenging for intra-dataset pretext tasks. For inter-dataset tasks, the model tends to learn to differentiate between datasets. We tackle this by ensuring that $\lambda$ is not simply predicted but the latent space corresponds to the time domain in terms of mixing distance. This ensures that we can really leverage cross-dataset information. Our experimental evaluation shows that this intuition carries over into practice.
>
> In initial probings, we did not see any meaningful effect by simply predicting the lambda as proposed by Wickstrøm et al. (2022).
>
> > The paper follows the same experimental setting as TF-C, however, reports a significantly lower performance and different metric than the reported one in the previous work.
>
> Yes, in favor of greater comparability, we followed the experimental setup of TF-C as an initial experiment in Tab. 1. Furthermore, this experiment illustrates that TF-C exhibits unstable training, prone to worsening or erratic behavior with each new dataset, leading to unpredictable performance. In contrast, XIT demonstrates robustness, with minimal decreases and significant increases across all cases. TS-TCC performs adequately with 5 datasets but lacks robustness, exhibiting similar instability to TF-C when more data is leveraged. In datasets like ECG, consistent increases are observed across all models, yet XIT benefits the most.
> This motivates our new scaling experiment, shown in Tab. 2 of the paper. A small excerpt is shown in the global response 2.
>
> We ran our experiments on the provided open-source code of TS-TCC as well as TF-C. However, the scores provided in the paper were not just reproducible. Also, note that we padded the time series to equal lengths, which was not performed in the respective initial publications. Similar difficulties were observed in the very extensive survey by Ma et al. (2023), see Table 13. There, the claimed superiority of TS-TCC could not be confirmed. Sadly, our attempts to contact the authors of TF-C were not responded to.
>
> > Table 1 reports an AUROC significantly lower than 0.5 for Epilepsy, which is a binary classification problem.
>
> This is indeed surprising, but not impossible. The model performs way worse than random guessing, which is the case when the model overfits on a single class, as an example. This is especially the case for imbalanced datasets.
>
> > The standard deviation of Table 2. is very high, which makes it hard to reason about the result.
>
> For this, we provided novel experiments (See answer 2).
>
> >  Moreover, there is no evidence provided that this would not apply to TS-TCC and it even seems to improve with more datasets.
>
> We provided more insights into the results of Tab. 1 in global answer 2. Furthermore, we ran additional experiments to show that TS-TCC does not scale to multiple datasets.
>
> >  Standard deviations in ablation
>
> Thank you for pointing this out. We improved the presentation of the results to make it easier to interpret them.See Global Response 3.
>
> >  What is the performance of XIT for Table 1 using the same setting as TS-TCC for their table 2 and fig 3?
>
> Our method makes inherent use of multiple datasets (specifically, through XD-MixUp and SICC). In contrast, Table 2 in Eldele et al. (2021) is a homogenous one-to-one experiment, e.g., HAR to HAR, which is not applicable in our case. Furthermore, their experiments assume an abundance of data, where supervised learning excels. We therefore evaluate a more challenging task, namely many-to-one training, e.g., in Tab. 1 and 2.
>
> We also report on the experiments of scaling dataset amounts (in our case, in the many-to-one setting) in the appendix Fig. 5. We went as low as forming a single batch was still possible. We can hereby confirm the intuitively convincing findings of TS-TCC that in the face of high finetuning data amounts, supervised learning becomes increasingly feasible.
>
> We hope that we have clarified your concerns and that you can reconsider our rating. Do let us know if you have any further questions or comments.

---

> ### Author Response · Authors · 2023-11-20
> **Looking Forward to Feedback on Our Response**
>
> Dear Reviewer,
>
> We appreciate the time and effort that you have taken to provide us with your insightful review. We would like to ask if the reviewer has any further concerns or is satisfied by our responses to the original review.
>
> We are looking forward to any further discussion with the reviewer and would like to thank the reviewer again for helping make our paper better.
>
> Regards,
> The Authors

---

### Author Response · Authors · 2023-11-16

Dear Reviewers,

We are really thankful for your in-depth review of our submission and welcome both the strengths you point out as well as your suggestions for improvements.

Here we address common aspects raised by more than one reviewer for a better overview.

We have already revised the paper that you can access with the key changes you kindly brought up. We will point out the relevant locations in the manuscript each time to make checking the improvements yourself easier.

### 1. Relevance and necessity of XIT

We would like to explain further why our method is not a small change to TS-TCC (Eldele et al., 2021) or just a trivial combination of existing methods. It, instead, is much more than the sum of its constituents and a necessary contribution to solving the challenges of multi-dataset training. First of all, as laid out in more depth in the related work (Section 4), no prior time series models attempted to tackle this challenge as pointed out by reviewer Gxh9. The paper on TF-C (Zhang et al., 2022) even spells out the common notion that this would be infeasible. With the privilege of hindsight, we can ascertain that this is a widespread misbelief. We substantiate this with two key experiments and demonstrate that our method is essential in solving this. Next, we evaluate more models as baselines on scaling to large portions of the UCR repository (Section 2. of this response), even adding previously missing ones. XIT clearly outperforms TS-TCC. Finally, in (Section 3. of this response), we lay out how we improved the ablation study and have improved the paper to present the insights more clearly.

### 2. Evaluate more models on scaling to 75 datasets and compare them directly

Some of you rightfully brought up the need to see that existing methods do not favorably scale to multiple datasets. We completely understand and agree with this concern. To show this, we have now run several new experiments:
- We ran the baseline models TS-TCC and TF-C and showed they do not benefit from adding more datasets as XIT does.
- We added more baselines, as you pointed out. Specifically, we now compare to the SOTA methods TS2Vec (Yue et al., 2022), TNC (Tonekaboni et al., 2020), and T-Loss (Franceschi et al., 2019).

The results when pretraining XIT and these *three new models* are shown below. Specifically, this is Macro F1 score across 25 UCR datasets not among the 75 pretrain datasets. The standard deviation is across 3 independent seeds. We added the mean and rank per model as the last two lines to better interpret the results while the complete table is on pg. 8 of the revised paper.

| Target Dataset          	| XIT           	| TS-TCC        	| TF-C           	| TS2Vec        	| TNC           	| T-Loss        	|
|-----------------------------|-------------------|-------------------|--------------------|-------------------|-------------------|-------------------|
| Average Macro F1 $\uparrow$ | **54.4±3.2** | 51.1±3.5      	| 43.1±6.1       	| 53.7±2.5      	| 36.4±9.1      	| 34.2±4.7      	|
| Rank $\downarrow$       	| **2.10±1.2**      	| 3.06±1.6      	| 3.56±1.8       	| 2.82±1.6      	| 4.48±1.0      	| 4.98±1.1 |

Note that the standard deviation for *Tiselac* is marked as `TBA` since that dataset is more extensive and requires more time to evaluate on TF-C. We wanted to notify you as soon as possible and will add that result later.

**Conclusion: One can clearly see that XIT outperforms all other methods with respect to the avg. score & rank (cf. the last two lines of the table). Additionally, it does so at moderate levels of variance.**

### 3. Significance of the ablation study

You are right in that the initial presentation of the ablation study was not very clear to interpret. The large standard deviation was an artifact of the evaluation, where we averaged over a diverse set of datasets with varying difficulty and, thereby, scores. We changed the table to show the ranks of the methods instead of the metrics directly, as recommended by Demšar (2006). The evaluation protocol is similar to the one on the UCR dataset, i.e., we pretrained on 75 UCR datasets and finetuned on a hold-out of 25 datasets. The results are as follows:

| Pretraining Component  	| AUROC rank $\downarrow$ | Accuracy rank $\downarrow$ | Macro F1 rank $\downarrow$ |
|----------------------------|-------------------------|----------------------------|----------------------------|
| XD-MixUp + SICC + TC (XIT) | **1.800 ±0.91**	| **1.780 ±0.89**   	| **1.780 ±0.84**   	|
| XD-MixUp + SICC        	| 3.560 ±0.87         	| 3.700 ±0.56            	| 3.580 ±0.79            	|
| XD-MixUp + TC          	| 2.060 ±0.94         	| 1.920 ±0.84            	| 1.980 ±0.91            	|
| TC                     	| 2.580 ±0.91         	| 2.600 ±0.78            	| 2.660 ±0.81            	|

**Conclusion: We improved the clarity of the ablation study to show that all three key components of XIT–namely XD-MixUp, TC, and SICC–are each relevant.**

---

### Author Response · Authors · 2023-11-21
**Request to engage in the discussion**

Dear reviewers,

Since the discussion period is coming to a close soon we would like to ask if the provided rebuttal has answered your concerns. To recap, we provide new experiments on pretraining on 75 datasets as well as several SOTA baselines. We also improve our ablation studies to more clearly show why our method is not just a simple extension of the previous methods. Everything is available in the revised version of the paper.

We hope to hear back from you.

Regards,

The Authors

---

### Meta-Review · Area_Chair_bA73 · 2023-12-06

**Metareview:**

The manuscript presents a pretraining method for time-series data sets, and is based on the TS-TCC. Although the paper lacks novelty, there is merit in the work. Summarizing the main reviewer comments below:
1. The novelity of the method is limited and the main components such as augmentations and TC loss are borrowed from previous works such as TS-TCC [Reviewer 4uSe)
2. The standard deviation in the ablation study is so large that it makes it hard to draw conclusions regarding the benefit of the components of the method. For instance one could argue that MixUp and TC are as good as XIT. (Reviewer KhAg)
3. Baselines are insufficient.

Summary: Although the paper lacks novelty, there is merit considering the volume of work (pertraining on 75 data sets). However, while doing this, the baselines are insufficient. Authors can work upon the reviews and submit an edited version to a different venue (Reviewer 4uSe).

**Justification For Why Not Higher Score:**

The manuscript has merits, but lacks novelty and/or sufficient baselines.

**Justification For Why Not Lower Score:**

N/A

---

### Decision · Program_Chairs · 2024-01-16

Reject